# The Cyclops Ophiolite as a Source of High-Cr Spinels from Marine Sediments on the Jayapura Regency Coast (New Guinea, Indonesia)

**Karol Zglinicki [1]** , **Krzysztof Szamałek [1,2,]\* and Irena Górska [1]**

1 Polish Geological Institute—National Research Institute, ul. Rakowiecka 4, 00-975 Warsaw, Poland; karol.zglinicki@pgi.gov.pl (K.Z.); irena.gorska@pgi.gov.pl (I.G.)
2 Faculty of Geology, University of Warsaw, ul. Zwirki i Wigury 93, 02-089 Warsaw, Poland
\* Correspondence: krzysztof.szamalek@uw.edu.pl

**Abstract:** The first detailed mineralogy, geochemistry and origin of heavy minerals in marine sediments along the Jayapura Regency coast on the Indonesian part of New Guinea Island are reported as part of a larger set of investigations conducted since 2009. In these sediments, the following heavy minerals were identified: high-Al and high-Cr spinels, chromian andradite, Mg-olivine, magnetite, mixture of iron (III) oxyhydroxides (limonite) and minerals from serpentine-group minerals (lizardite, antigorite). The heavy mineral fraction of marine sediments contains increased concentrations of metals, including W (up to 257.72 ppm) and Ag (up to 1330.29 ppb) as well as minor amounts of Ni (7.1–3560.9 ppm) and Cr (68.0–5816.0 ppm). The present state of geological knowledge suggests that there are no known prospects for rich Ti, Ni, Co, Cr, Au deposits along the examined part of the Jayapura coast. However, the average content of Ag and W is high enough to provide an impulse for suggested further deposit research. The source of marine sediments is Cyclops ophiolite, which contains a typical ophiolite sequence. Cyclops Mountain rocks have undergone intense chemical weathering processes and the resulting eroded material has been deposited on the narrow continental shelf. The chemical composition of chromian spinels indicates that their source is depleted peridotites from the SSZ (supra-subduction zone) environment of the Cyclops ophiolite. A detailed geochemical examination indicates that the evolution of parental melt of these rocks evolved towards magma with geochemical parameters similar to mid-ocean ridge basalt (MORB).

**Keywords:** chromian spinels; cyclops ophiolite; chromian garnet; New Guinea Island; heavy minerals; placer deposits

## 1. Introduction

The geological evolution of SE Asia and New Guinea is recognized as being highly complex. Collisions of the Indo-Australian Plate with the Eurasian Plate and the formation of many subduction zones with related extension processes, such as the opening of pull-apart basins, had the strongest impact in the region [1–3]. Tectonic collisions and closing of marginal basins in the Cenozoic resulted in the formation of numerous ophiolites [4] in SE Asia and the Pacific region, *i.e.*, the Cyclops ophiolite [5], Papuan Ultramafic Belt [6,7], Seram-Ambon ophiolite [8] and ophiolites in the Balantak and central Sulawesi regions [9].

Ophiolites from SE Asia may be split into two groups. The first group is autochthonous ophiolites connected with the Eurasian continent [4]. It reflects the closing of marginal basins, back-arc basins and island arcs adjacent to the continent. The formation of these ophiolites related to the closing of the Tethys Ocean, the proto-South China Sea and the obduction of seafloor fragments from marginal

basins [4], including ophiolites associated with the Woyla Group in Sumatra [10,11], the Meratus ophiolite in SE Borneo [12] and the northern and eastern segments of Sulawesi Island [9].

The second group of ophiolites located on New Guinea (e.g., the Cyclops ophiolite and the Central Ophiolite Belt) is suggested to have a supra-subduction origin. They contain fragments from the Philippine Sea basement and the Pacific Plate [4,5,13]. These fragments were obducted from the north on the northern edge of the Indo-Australian Plate [4,5,13].

Numerous metal deposits of economic interest are associated with ophiolite sequences in SE Asia and the SW Pacific region, *i.e.*, podiform-type chrome deposits—Luzon, Palawan Philippines [14]; Ni-Co weathering deposits—Sulawesi, Halmahera, Indonesia [15,16]; and platinum group minerals (PGMs)—New Caledonia, France [17]. Research carried out in Waropen Regency, New Guinea (Figure 1) revealed high chromium concentrations of economic significance in alluvial sediments [18]. The origin of chromian spinels from the Waropen Regency is most likely tied to the petrological ophiolite sequence similar to the Cyclops ophiolite [18].

Cyclops Mountain rocks underwent intense chemical weathering processes. Eroded material is deposited on the narrow continental shelf at its edge—The New Guinea Trench [19,20]. Detailed mineralogical and chemical characteristics of the marine sediments are given by [19]. Sediments from the narrow shelf at the base of the Cyclops Mountains may possess economic potential. The Department of Mineral and Energy Resources in the Papua Province selected six geological provinces with high deposit potential within the Indonesian part of New Guinea. These provinces are: Western Irian Jaya, Cenderwasih Bay, Jayapura, Central, Southern Papua and Fafak. The Jayapura geological province contains the Cu–Zn–Pb–Fe–Cr–Co–Ni–Au mineralization zone. Exploration work conducted since the 1920s in the Jayapura area in the following locations: Tanahmerah, Tablasufa, Rhynauwen, Amaybu, Kirpon, Harpan, Dojo [21–23], led to the discovery of Ni-Co laterite deposits by the Pacific Nikkel Company. This company's reports indicated resources to be as high as 44.3 million tons [24,25], with average concentrations of Ni being 1.31% and Co 0.11%, with a 0.80% nickel cutoff. In the Tanahmerah Region, platinum and palladium were found and documented (Pt up to 35 ppb, Pd up to 10 ppb) [25]. Predicted and undiscovered deposits may exist around the Cyclops Mountains, including alluvial deposits of Ag-Au and marine Ti, Ni, Co, Cr, Au, Ag, PGE and Au epithermal placer deposits on the coastline. The high deposit potential of New Guinea and the presence of already discovered deposits are reasons for suggested further and more detailed exploration work.

Since 2009, a team of Polish geologists from the University of Warsaw and the Polish Geological Institute—National Research Institute have conducted geological prospective studies in New Guinea, including the Jayapura area [18,19]. For the first time they described the petrological and mineralogical composition of marine sediments around the Cyclops Mountains [19]. This article presents results of detailed analyses of heavy minerals separated from sediments containing chromian spinels. The goal of this research was to determine the detailed chemical composition of chromian spinels and their origin, as well as to assess the deposit potential of the heavy mineral fraction.

## 2. Geological Settings

The Cyclops Mountains are an isolated block along the coast in the northern part of New Guinea Island (Figure 1). The mountains may be considered a complex anticline represented by two slices thrust over an unknown basement [5]. The anticline's core is composed of metamorphic rocks formed during the ophiolite obduction. The upper part of the geological profile consists of ophiolite sequence rocks. Locally, the southern part of the massif is covered by Tertiary volcano-clastic sediments of the Auwewa Volcanic Group, sheeted limestones of the Hollandia Group and contemporaneous sands and gravels [5,21,26]. The geological setting of the Cyclops ophiolite is typical for an ophiolite series. It is composed of residual peridotites from the Earth's mantle, cumulates of isotropic gabbros, massive dolerites, troctolite and various lavas, including pillow lavas [4,5,26].

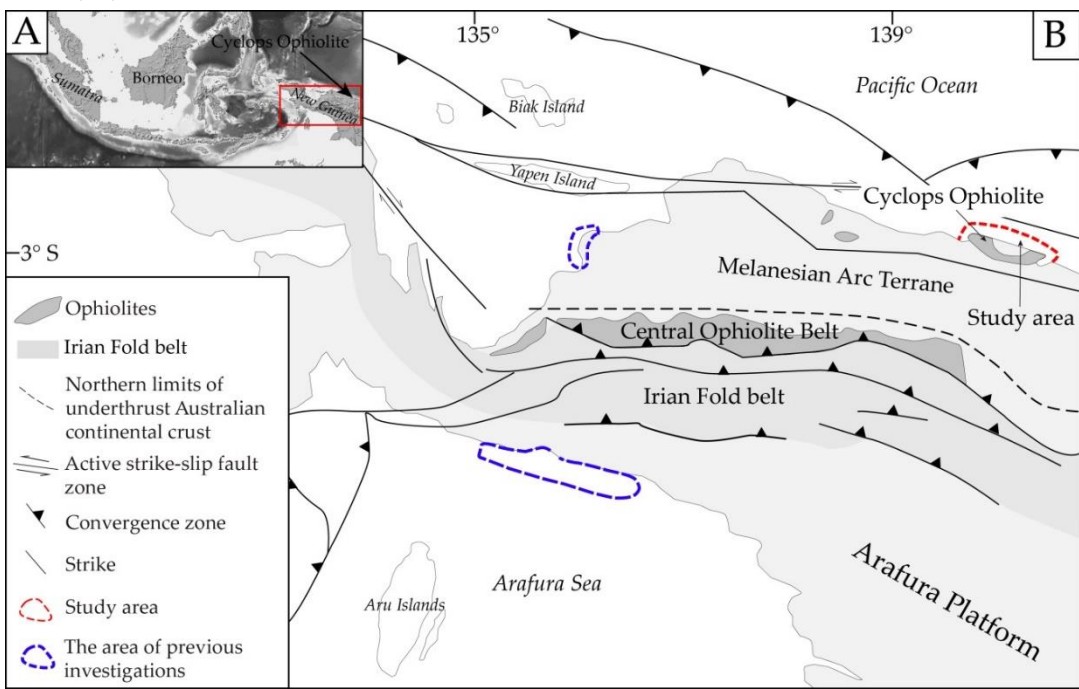

**Figure 1.** (**A**) Map of the Malay Archipelago; (**B**) Tectonic sketch of New Guinea Island with marked exploration area. Modified from [13,18,19,27,28].

## 3. Materials and Methods

A total of 69 samples of sediments were collected from beaches and shallow seabed on the northern coast of New Guinea (Jayapura area). A detailed description of the sampling methods was provided by [19,29]. The heavy mineral fraction (0.50 to 0.063 mm in diameter) obtained from bulk samples of marine sediments (Figure 2A,B) was separated using heavy liquid (sodium polytungstate, density $2.89 \pm 0.02$ g/cm$^3$).

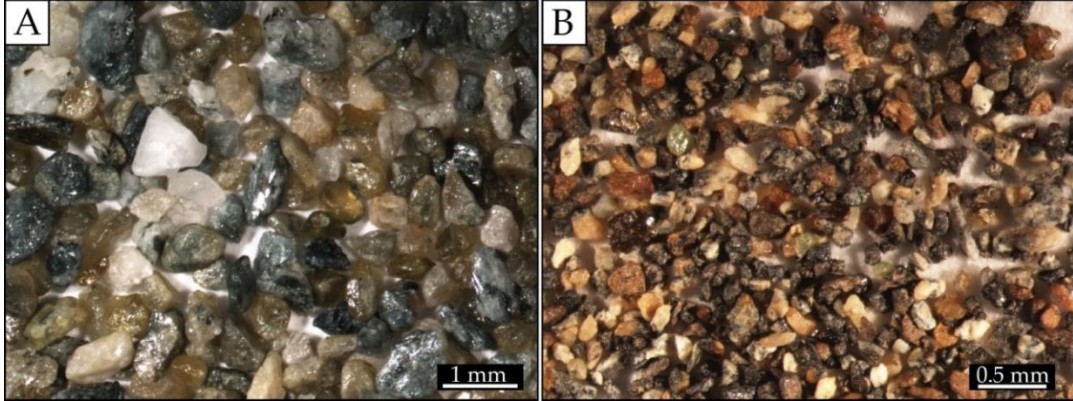

**Figure 2.** Stereoscopic microscope images. (**A**) Marine sediments from the Jayapura area. Bulk sample JP49, magnification ×2; (**B**) heavy minerals separated from sample JP07, magnification ×0.8.

Polished thin sections (150 plates) of the heavy mineral concentrate were examined. Chromian spinels, Cr-garnets and olivine were analyzed to characterize their microstructures, base metals and inclusions using a scanning electron microscope (SEM) and electron probe microanalyzer (EPMA). The minerals were observed using the scanning electron microprobe FE-SIGMA VP (Carl Zeiss Microscopy Ltd., Cambridge, UK) equipped with two detectors EDS (SDD XFlash| 10) (Bruker, Berlin, Germany). Analytical conditions were accelerating voltage of 25 kV in a high vacuum. Chemical

compositions of minerals were obtained with a Cameca SXFiveFe (Cameca, Gennevilliers, France) electron microprobe at the University of Warsaw, Warsaw, Poland. The electron microprobe is equipped with wave-length dispersive spectrometers (WDS) with an acceleration voltage of 15 kV and a beam current of 20 nA, 20 s counting time, 10 s background time, and beam size of several nm. The analyses were performed on TAP (thallium acid phthalate), PET (pentaerythritol), LPET(large pentaerythritol) and LLIF(large lithium fluoride) crystals (Table 1).

**Table 1.** Electron probe microanalyzer (EPMA) calibration standards.

| Element | Standard | Analytical Line | Crystal | Detection Limit (wt%) |
|---------|----------|-----------------|---------|-----------------------|
| Na | Albite | K$\alpha$ | TAP | 0.02–0.04 |
| Mg | Diopside | K$\alpha$ | TAP | 0.02 |
| Si | Diopside | K$\alpha$ | TAP | 0.02 |
| Al | Dysten | K$\alpha$ | TAP | 0.02–0.03 |
| Fe | $Fe_2O_3$ | K$\alpha$ | LIF | 0.08–0.12 |
| Mn | Rhodonite | K$\alpha$ | LIF | 0.08–0.11 |
| Ca | Diopside | K$\alpha$ | PET | 0.03–0.04 |
| V | $V_2O_5$ | K$\alpha$ | LIF | 0.06–0.08 |
| Co | CoO | K$\alpha$ | LIF | 0.08–0.10 |
| Ni | NiO | K$\alpha$ | LIF | 0.11–0.15 |
| Cu | Cuprite | K$\alpha$ | LIF | 0.14–0.17 |
| Cr | Chromite | K$\alpha$ | PET | 0.05–0.06 |
| Zn | Sphalerite | K$\alpha$ | LIF | 0.18–0.23 |
| Ti | Rutile | K$\alpha$ | PET | 0.03–0.05 |
| Sc | Pure Sc | K$\alpha$ | LPET | 0.01–0.02 |
| Ga | GaAs | L$\alpha$ | TAP | 0.10–0.13 |

EPMA calibration was performed using standards (Table 1) included in equipment of the Inter-Institutional Laboratory of Microanalysis of Minerals and Synthetic Substances at the Faculty of Geology, University of Warsaw. Results were corrected on the basis of a CAMECA PAP algorithm by Pouchou and Pichoir [30].

The phase composition was investigated using an AXS D8 Advance Davinci Bruker diffractometer equipped with a copper anode lamp. Diffractograms were recorded in the angle range of 3–85° 2θ (Cu K$\alpha$), measurement step 0.02°, and measurement time: 2.5 s/step. Crystal line phases were identified using X'Pert HighScore Plus software by comparing the registered diffractograms with ICDD PDF-2 and PDF-4+ standards.

The chemical composition of bulk samples and selected fractions (<0.50 mm; <0.25 mm, <0.10 mm) from heavy mineral concentrates was determined by the Bureau Veritas (BV) certified laboratory in Vancouver (BC, Canada). The samples were analyzed using LF202 and AQ250 analytical programs (www.acmelab.com). The lower and upper detection limit is correlated with BV analytical programs. The samples were dissolved (analytical program AQ250) in a mixture of aggressive acids (1:1:1 $HNO_3$:HCl:$H_2O$). Samples were melted using program LF202 (with $Na_2B_4O_7$/$Li_2B_4O$) and then dissolved in a mixture of aggressive acids. In bulk samples, only Pt and Pd (from among PGEs) were analyzed. The analyses were carried out using inductively coupled plasma mass spectrometry (ICP-MS) and inductively coupled plasma optical emission spectroscopy (ICP-OES) (Table 1).

## 4. Results

### 4.1. Heavy Minerals Concentrate

The heavy mineral fraction of the seafloor sediments is differentiated laterally along the coastline. The area of the Jayapura coast was divided into three Units: Western, Central and Eastern on the basis of a mineralogical and chemical analysis [19,29]. The average content of heavy minerals changes linearly from the East (1.93 wt%) to West (24.92 wt%). In the Central Unit the average content of heavy

fraction is 26.47 wt%. In coastal sediments, a linear decrease in heavy mineral content with increasing water depth is evident. Visible digression of the heavy fraction is below the isobath level of 27 m below the seafloor (m b.s.f.). The share of heavy minerals in the sediment by weight is 45.76 wt%, whereas at an isobath of 30 m b.s.f. it is only 5.73 wt%. The greatest amount of heavy minerals (average 51.96 wt%) is recorded at depths between 23 and 27 m b.s.f. The heterogeneity of the sediments also appears to be related to distance from land. In fact, the average content of heavy minerals decreases linearly from the coast (48.33 wt%) to a distance of 460 m from land (0.21 wt%). The greatest share of the heavy fraction in seabed sediments occurs at distances between 200 and 300 m from the coast. Sediments of all units contain heavy minerals (dominating in grain class 0.10–0.25 mm), which are poorly rounded. They often form paragenesis, among others: chromian spinel with olivine, amphibole with epidote, Cr-garnet with olivine. Additionally, multiple exsolutions and forms of replacement in mineral grains have been observed. Ilmenites form characteristic exsoluted differentiated structures in hematite. These structures occur as lamellae, drops, lens and rarely irregular forms. Ilmenite replacement leads to formation of second generation of hematite and is quite often rutile. Visible structures of ilmenite replacement and/or chemical changes cause transformation of ilmenite to titanite. The heavy fraction from Eastern and Western Units is composed of chromian spinels (Figure 3A,B), olivine, serpentine subgroup, Cr-garnets, magnetite, a small amount of pyroxene and a mixture of hydrated iron (III) oxide-hydroxides (limonite). Heavy fraction from the Eastern Unit is composed of homogenous chromite grains (7.36%), whereas chromite in olivine-magnetite paragenesis is as low as 1.44% of the heavy fraction. The Western Unit is characterized by average chromite grain content at 2.71% with minor magnetite (0.30%) and olivine (0.79%). The Central Unit consists of amphiboles, epidotes, clinochlore, pyroxene, siderite, and minor amounts of ore minerals: ilmenite, hematite, magnetite, rutile, pyrite and chalcopyrite (Figure 4A,B). In the Central Unit, the homogenous chromite grains content is on average 4.98% in paragenesis with magnetite (1.22%) and olivine (0.02%).

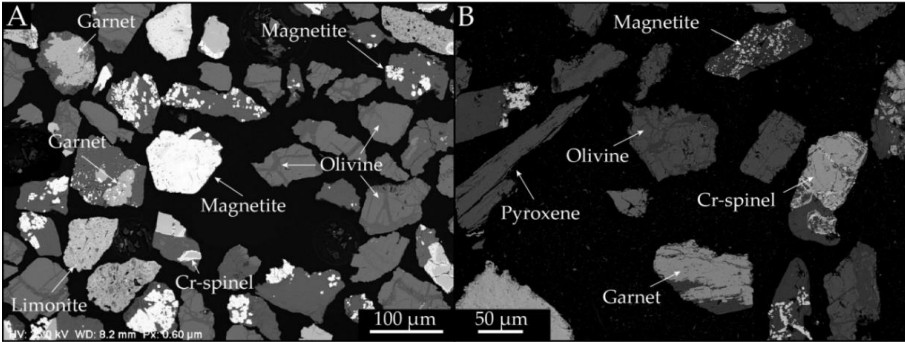

**Figure 3.** (**A**) The heavy minerals fraction from sample JP 08 (fraction < 0.1 mm); (**B**) the heavy minerals fraction from sample JP 07 (fraction < 0.25 mm). SEM-BSE images.

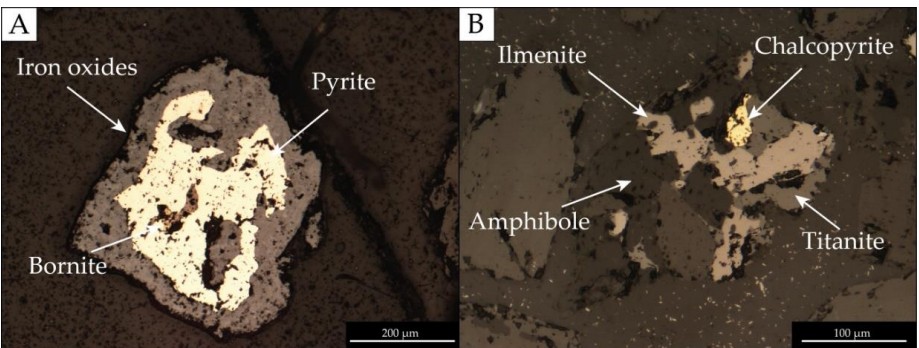

**Figure 4.** Sulphides in the Central Unit sediments. Sample JP36. (**A**) Pyrite dispersed in iron oxides; (**B**) chalcopyrite within amphibolite. Replacement of ilmenite with titanite. Reflected-light microscopy images. Magnification ×5.

The chemical composition of sediments (Table 2) shows a spatial differentiation in the content of metals. Most interesting are the content of Ni (between 7.1–3560.9 ppm; on average 972.55 ppm), Cr (68.0–5816.0 ppm; on average 1962.63 ppm), W (<0.5–3130.3; on average 257.72 ppm) and Ag (0.1–5000.00 ppb; on average 1330.29 ppb). The content of metals in the heavy fraction is similar to bulk samples (Appendix A). The heavy minerals concentrate contains Ag in a higher amount than in bulk samples.

### 4.2. Chromian Spinels

Chromian spinels are the main ore minerals found in the beach and seabed sediments. Spinel grains usually form allotriomorphic minerals and occur in all grain fractions (0.50 to 0.063 mm). Different chromian spinels were identified (using optical microscopy and SEM—About 500 analyses): (a) chromian spinel coexisting in paragenesis with magnetite and serpentinized olivine (Figure 5A); (b) homogenous grains of chromian spinel (Figure 5B); (c) altered, metamorphic grains of chromian spinel (Figure 5B,C). EPMA analyses show significantly different compositions on the Cr-Al-$Fe^{3+}$ ternary diagrams (Figure 6A). Chromian spinels form a continuous magnesiochromite—Chromite series (Figure 6B).

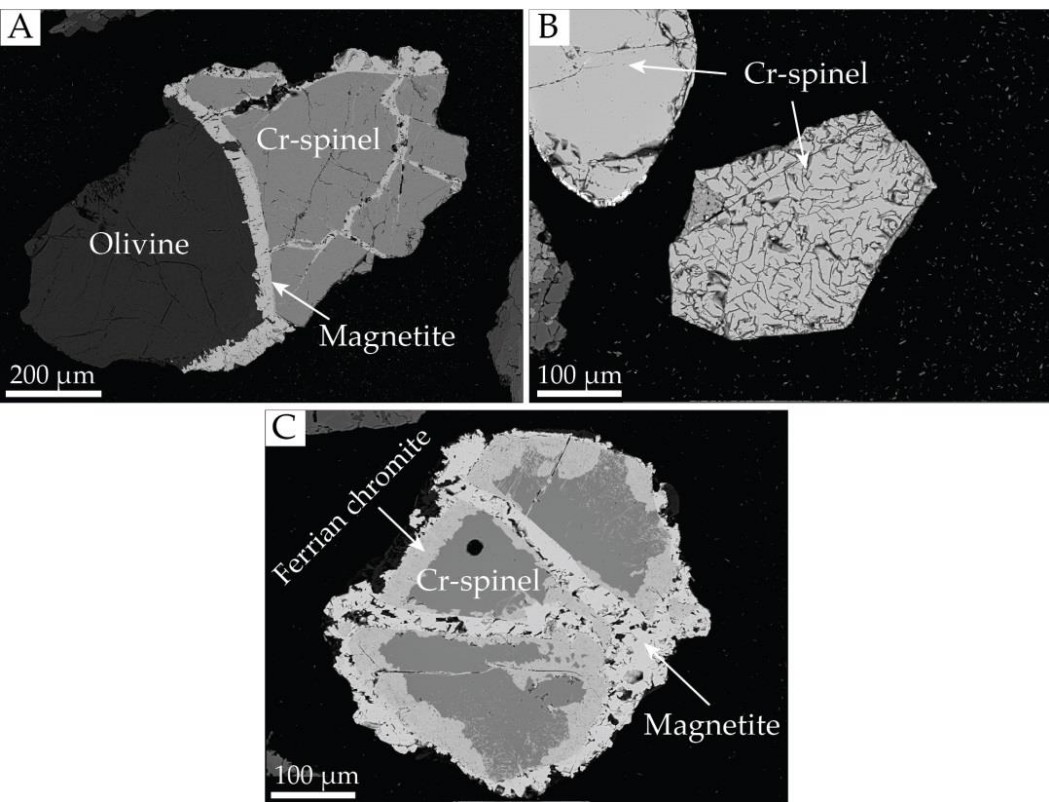

**Figure 5.** (**A**) Chromian spinel in paragenesis with magnetite and serpentinized olivine (sample JP 08); (**B**) homogenous grains of chromian spinel (sample JP 10); (**B**,**C**) altered grains of chromian spinel (sample JP 13). SEM-BSE images.

The Cr# and Mg# for chromium phases are characteristic for continuous series: magnesiochromite—chromite. Most of the grains are relatively rich in Cr, whereas others are Al-enriched. Based on the differences of chemical composition described by the Cr# = [100·Cr/(Cr + Al)], chromian spinels can be divided into two main groups [31]: high-Al spinel (with Cr# < 60) and high-Cr spinel (with Cr# > 60). Their composition is reported in Table 3.

**Table 2.** The chemical composition of sediments (inductively coupled plasma mass spectrometry (ICP-MS) method, in ppm).

| Element | Western Unit | | | | | | | Central Unit | | | | | | Eastern Unit | | | | | |
|---------|------|------|------|------|------|------|------|------|------|------|------|------|------|------|------|------|------|------|------|
| | JP36 | JP38 | JP40 | JP41 | JP43 | JP49 | JP53 | JP13 | JP14 | JP15 | JP29 | JP32 | JP4 | JP5 | JP6 | JP7 | JP8 | JP9 | JP10 |
| Cu | 18.40 | 7.26 | 11.92 | 12.04 | 6.77 | 2.90 | 39.4 | 13.13 | 17.27 | 12.01 | 27.98 | 13.67 | 9.20 | 5.50 | 2.22 | 2.61 | 2.00 | 1.90 | 2.28 |
| Cr | 274.0 | 274.0 | 274.0 | 68.0 | 68.0 | 2532.0 | 274.0 | 4721.0 | 137.0 | 68.0 | 274.0 | 137.0 | 5337.0 | 5816.0 | 2463.0 | 3421.0 | 4310.0 | 4653.0 | 2189.0 |
| Ni | 56.0 | 107.0 | 42.0 | 31.0 | 7.1 | 2075.0 | 74.5 | 724.0 | 30.0 | 27.0 | 84.0 | 28.0 | 3560.9 | 2744.0 | 1217.8 | 1922.2 | 2033.0 | 2055.0 | 1660.0 |
| Pb | 0.54 | 0.36 | 0.50 | 0.46 | 0.52 | 0.18 | 0.60 | 0.36 | 0.37 | 0.49 | 0.36 | 0.31 | 5.60 | 7.60 | 0.60 | 0.24 | 0.57 | 0.55 | 0.32 |
| Zn | 31.4 | 8.1 | 19.9 | 45.6 | 18.2 | 21.5 | 54.0 | 22.9 | 12.4 | 14.7 | 32.4 | 19.5 | 36.0 | 27.0 | 14.0 | 22.7 | 27.0 | 27.3 | 21.8 |
| Co | 6.6 | 5.8 | 4.2 | 6.7 | 3.7 | 63.7 | 19.6 | 35.2 | 6.5 | 7.1 | 9.5 | 4.6 | 141.7 | 117.6 | 47.4 | 68.0 | 82.0 | 82.2 | 62.3 |
| Cd | 0.23 | 0.02 | 0.13 | 0.20 | 0.03 | 0.07 | <0.01 | 0.05 | 0.07 | 0.02 | 0.13 | <0.01 | <0.01 | <0.01 | 0.03 | 0.10 | 0.06 | 0.06 | 0.03 |
| Sb | 0.17 | <0.02 | <0.02 | 0.02 | 0.06 | 0.02 | 0.02 | 0.04 | 0.06 | 0.04 | 0.02 | <0.02 | 0.02 | 0.02 | 0.06 | 0.03 | 0.03 | 0.03 | 0.04 |
| Bi | <0.02 | <0.02 | <0.02 | <0.02 | <0.02 | <0.02 | <0.02 | <0.02 | <0.02 | <0.02 | <0.02 | <0.02 | <0.02 | <0.02 | <0.02 | <0.02 | <0.02 | <0.02 | <0.02 |
| Mo | 0.05 | 0.18 | 0.05 | 0.07 | 0.10 | 0.08 | 0.70 | 0.19 | 0.17 | 0.20 | 0.08 | 0.06 | 0.60 | 0.50 | 0.20 | 0.30 | 0.15 | 0.15 | 0.12 |
| Sn | <1 | <1 | 2 | <1 | <1 | <1 | <1 | <1 | <1 | <1 | <1 | <1 | <1 | <1 | <1 | <1 | <1 | <1 | <1 |
| V | 137.0 | 41.0 | 175.0 | 119.0 | 118.0 | 29.0 | 146.0 | 215.0 | 244.0 | 75.0 | 118.0 | 62.0 | 63.0 | 77.0 | 22.0 | 24.0 | 33.0 | 36.0 | 27.0 |
| W | <0.5 | <0.5 | <0.5 | <0.5 | <0.5 | <0.5 | 173.6 | <0.5 | <0.5 | <0.5 | <0.5 | <0.5 | 3130.3 | 1592.9 | <0.5 | <0.5 | <0.5 | <0.5 | <0.5 |
| Zr | 53.6 | 11.7 | 51.6 | 65.1 | 39.9 | 0.9 | 51.9 | 47.8 | 21.5 | 20.9 | 46.3 | 54.8 | 2.5 | 3.9 | 1.2 | 0.9 | 2.2 | 2.6 | 3.1 |
| Nb | 0.1 | <0.1 | <0.1 | 0.2 | <0.1 | <0.1 | 0.8 | 3.0 | 0.7 | 0.2 | 0.3 | 0.4 | 0.8 | <0.1 | <0.1 | 0.3 | 0.2 | <0.1 | <0.1 |
| Hf | 1.6 | 0.3 | 1.6 | 2.0 | 1.1 | <0.1 | 1.6 | 1.1 | 0.6 | 0.6 | 1.3 | 1.6 | <0.1 | 0.1 | <0.1 | <0.1 | <0.1 | <0.1 | <0.1 |
| Th | 0.7 | <0.2 | 1.1 | 0.6 | 0.4 | <0.2 | 0.3 | 0.5 | 0.5 | 0.3 | 0.3 | 0.4 | <0.2 | <0.2 | <0.2 | <0.2 | <0.2 | <0.2 | <0.2 |
| U | 0.4 | 0.8 | 0.5 | 0.7 | 1.0 | 0.6 | 1.0 | 0.5 | 1.6 | 1.5 | 0.3 | 0.4 | 0.3 | 0.4 | 1.3 | 0.6 | 0.5 | 0.5 | 0.8 |
| | | | | | | | | ppb | | | | | | | | | | | |
| Hg | 12.0 | 6.0 | 12.0 | 16.0 | 8.0 | 16.0 | <5 | 15.0 | 18.0 | 15.0 | 12.0 | <5 | <5 | <5 | 19.0 | 17.0 | 16.0 | 19.0 | 17.0 |
| Ag | 3408.0 | 4.0 | 2832.0 | 3736.0 | 5.0 | 1139.0 | 3.0 | 1008.0 | 865.0 | 8.0 | 2660.0 | 5000.0 | 0.1 | 0.4 | 215.0 | 1825.0 | 905.0 | 1268.0 | 394.0 |
| Au | 0.6 | 3.6 | 0.4 | 0.3 | 1.8 | <0.2 | 1.5 | 4.0 | <0.2 | 0.2 | 0.4 | 0.4 | 3.6 | 2.7 | <0.2 | <0.2 | 3.0 | 4.0 | 2.0 |
| Pt | <3 | <3 | <3 | <3 | <3 | 9.0 | <3 | <3 | <3 | <3 | <3 | <3 | 4.0 | 5.0 | 6.0 | 4.0 | 8.0 | <3 | <3 |
| Pd | <10 | <10 | <10 | <10 | <10 | <10 | <10 | <10 | <10 | <10 | <10 | <10 | <10 | <10 | <10 | <10 | <10 | <10 | <10 |

**Table 3.** The chemical composition of high-Cr and high-Al spinels (in wt%) with EPMA.

| High-Cr Spinel | | | | | | | | | | | | | | | | | |
|---|---|---|---|---|---|---|---|---|---|---|---|---|---|---|---|---|---|
| Sample | JP49-1 | JP49-2 | JP49-3 | JP49-4 | JP49-5 | JP49-6 | JP09-1 | JP09-2 | JP09-3 | JP09-4 | JP07-1 | JP07-2 | JP07-3 | JP07-4 | JP13-1 | JP13-2 | JP13-3 |
| $SiO_2$ | 0.06 | 0.09 | 0.05 | 0.09 | 0.07 | 0.06 | 0.05 | 0.04 | 0.04 | 0.03 | 0.06 | 0.06 | 0.04 | 0.04 | 0.08 | 0.05 | 0.08 |
| $TiO_2$ | 0.05 | 0.04 | 0.24 | 0.24 | 0.27 | 0.18 | 0.00 | 0.03 | 0.08 | 0.08 | 0.00 | 0.02 | 0.33 | 0.30 | 0.11 | 0.12 | 0.15 |
| $Al_2O_3$ | 18.94 | 17.26 | 18.79 | 18.64 | 19.07 | 14.94 | 14.87 | 15.39 | 15.55 | 15.31 | 21.42 | 21.18 | 15.88 | 15.33 | 19.63 | 20.56 | 20.56 |
| $Fe_2O_3$ | 7.22 | 9.38 | 2.01 | 1.32 | 1.64 | 2.72 | 2.93 | 2.54 | 3.01 | 2.96 | 0.03 | 0.10 | 4.45 | 7.00 | 3.19 | 3.05 | 2.63 |
| $Cr_2O_3$ | 42.70 | 42.03 | 49.11 | 50.20 | 49.44 | 51.19 | 52.58 | 52.63 | 51.79 | 51.74 | 49.36 | 48.80 | 49.10 | 47.48 | 45.94 | 45.75 | 45.17 |
| $V_2O_3$ | 0.18 | 0.22 | 0.19 | 0.21 | 0.15 | 0.13 | 0.26 | 0.22 | 0.18 | 0.26 | 0.14 | 0.18 | 0.24 | 0.14 | 0.22 | 0.17 | 0.11 |
| FeO | 22.14 | 23.06 | 17.26 | 17.55 | 17.55 | 22.49 | 19.10 | 18.99 | 18.49 | 18.79 | 17.05 | 17.30 | 19.93 | 21.05 | 20.72 | 21.38 | 21.13 |
| MgO | 7.99 | 7.23 | 11.54 | 11.40 | 11.50 | 7.58 | 9.83 | 9.97 | 10.28 | 9.96 | 11.93 | 11.71 | 9.35 | 8.83 | 9.09 | 8.90 | 8.72 |
| MnO | 0.08 | 0.10 | 0.00 | 0.00 | 0.00 | 0.00 | 0.00 | 0.00 | 0.00 | 0.00 | 0.00 | 0.00 | 0.06 | 0.14 | 0.04 | 0.08 | 0.15 |
| CoO | 0.08 | 0.03 | 0.07 | 0.08 | 0.07 | 0.11 | 0.11 | 0.11 | 0.12 | 0.08 | 0.10 | 0.03 | 0.14 | 0.00 | 0.14 | 0.16 | 0.07 |
| CuO | 0.02 | 0.06 | 0.00 | 0.10 | 0.08 | 0.05 | 0.00 | 0.05 | 0.00 | 0.00 | 0.00 | 0.02 | 0.06 | 0.00 | 0.00 | 0.02 | 0.01 |
| ZnO | 0.43 | 0.36 | 0.06 | 0.12 | 0.00 | 0.29 | 0.15 | 0.19 | 0.18 | 0.26 | 0.23 | 0.03 | 0.32 | 0.09 | 0.37 | 0.42 | 0.58 |
| NiO | 0.12 | 0.13 | 0.06 | 0.13 | 0.13 | 0.09 | 0.09 | 0.08 | 0.07 | 0.11 | 0.04 | 0.09 | 0.14 | 0.11 | 0.07 | 0.05 | 0.05 |
| Total | 100.01 | 99.99 | 99.38 | 100.08 | 99.97 | 99.83 | 99.97 | 100.24 | 99.79 | 99.58 | 100.36 | 99.52 | 100.04 | 100.51 | 99.60 | 100.71 | 99.41 |
| Cr# | 60 | 62 | 64 | 64 | 63 | 70 | 70 | 70 | 69 | 69 | 61 | 61 | 67 | 68 | 61 | 60 | 60 |
| Mg# | 39 | 36 | 54 | 54 | 54 | 38 | 48 | 48 | 50 | 49 | 56 | 55 | 46 | 43 | 44 | 43 | 42 |
| Cr/Fe | 1.41 | 1.26 | 2.43 | 2.53 | 2.46 | 1.94 | 2.29 | 2.34 | 2.31 | 2.28 | 2.73 | 2.65 | 1.94 | 1.64 | 1.84 | 1.79 | 1.82 |
| Estimated composition for parental magmas | | | | | | | | | | | | | | | | | |
| $Al_2O_{3\ melt}$ * | 14.24 | 13.76 | 14.20 | 14.16 | 14.82 | 13.00 | 12.98 | 13.16 | 13.21 | 13.13 | 14.89 | 14.83 | 13.32 | 13.14 | 14.43 | 14.67 | 14.67 |
| $TiO_{2\ melt}$ * | 0.14 | 0.13 | 0.35 | 0.35 | 0.38 | 0.28 | 0.09 | 0.12 | 0.18 | 0.17 | 0.09 | 0.11 | 0.45 | 0.41 | 0.21 | 0.22 | 0.25 |
| $FeO/MgO_{melt}$ * | 0.88 | 0.98 | 0.29 | 0.32 | 0.33 | 0.91 | 0.48 | 0.47 | 0.41 | 0.46 | 0.31 | 0.34 | 0.58 | 0.66 | 0.53 | 0.59 | 0.60 |
| Formula, calculated on the basis of 32 oxygens (a.p.f.u.) | | | | | | | | | | | | | | | | | |
| Si | 0.02 | 0.02 | 0.01 | 0.02 | 0.02 | 0.02 | 0.01 | 0.01 | 0.01 | 0.01 | 0.01 | 0.01 | 0.01 | 0.01 | 0.02 | 0.01 | 0.02 |
| Ti | 0.01 | 0.01 | 0.05 | 0.05 | 0.05 | 0.03 | 0.00 | 0.01 | 0.02 | 0.02 | 0.00 | 0.00 | 0.06 | 0.06 | 0.02 | 0.02 | 0.03 |
| Al | 5.77 | 5.33 | 5.61 | 5.55 | 5.67 | 4.65 | 4.55 | 4.68 | 4.74 | 4.69 | 6.26 | 6.25 | 4.86 | 4.70 | 5.94 | 6.14 | 6.21 |
| $Fe^{3+}$ | 1.41 | 1.85 | 0.38 | 0.25 | 0.31 | 0.54 | 0.57 | 0.50 | 0.59 | 0.58 | 0.01 | 0.02 | 0.87 | 1.37 | 0.62 | 0.58 | 0.51 |
| Cr | 8.73 | 8.71 | 9.85 | 10.02 | 9.86 | 10.68 | 10.79 | 10.74 | 10.58 | 10.63 | 9.68 | 9.66 | 10.08 | 9.77 | 9.32 | 9.17 | 9.16 |
| V | 0.04 | 0.05 | 0.04 | 0.04 | 0.03 | 0.03 | 0.05 | 0.05 | 0.04 | 0.05 | 0.03 | 0.04 | 0.05 | 0.03 | 0.04 | 0.03 | 0.02 |
| $Fe^{2+}$ | 4.79 | 5.06 | 3.66 | 3.71 | 3.70 | 4.97 | 4.15 | 4.10 | 4.00 | 4.08 | 3.54 | 3.62 | 4.33 | 4.58 | 4.45 | 4.53 | 4.53 |
| Mg | 3.08 | 2.82 | 4.36 | 4.29 | 4.32 | 2.98 | 3.81 | 3.84 | 3.96 | 3.86 | 4.41 | 4.37 | 3.62 | 3.42 | 3.48 | 3.36 | 3.33 |
| Mn | 0.02 | 0.02 | 0.00 | 0.00 | 0.00 | 0.00 | 0.00 | 0.00 | 0.00 | 0.00 | 0.00 | 0.00 | 0.01 | 0.03 | 0.01 | 0.02 | 0.03 |
| Co | 0.02 | 0.01 | 0.01 | 0.02 | 0.01 | 0.02 | 0.02 | 0.02 | 0.03 | 0.02 | 0.02 | 0.01 | 0.03 | 0.00 | 0.03 | 0.03 | 0.01 |
| Cu | 0.00 | 0.01 | 0.00 | 0.02 | 0.02 | 0.01 | 0.00 | 0.01 | 0.00 | 0.00 | 0.00 | 0.00 | 0.01 | 0.00 | 0.00 | 0.00 | 0.00 |
| Zn | 0.08 | 0.07 | 0.01 | 0.02 | 0.00 | 0.06 | 0.03 | 0.04 | 0.04 | 0.05 | 0.04 | 0.01 | 0.06 | 0.02 | 0.07 | 0.08 | 0.11 |
| Ni | 0.01 | 0.01 | 0.01 | 0.01 | 0.01 | 0.01 | 0.01 | 0.01 | 0.01 | 0.01 | 0.00 | 0.01 | 0.01 | 0.01 | 0.01 | 0.00 | 0.00 |

**Table 3.** *Cont.*

| Sample | JP11-1 | JP11-2 | JP11-3 | JP06-1 | JP06-2 | JP06-3 | JP06-4 | JP07-5 | JP07-6 | JP07-7 | JP07-8 | JP14-1 | JP14-2 | JP14-3 | JP14-4 | JP14-5 | JP14-6 |
|---|---|---|---|---|---|---|---|---|---|---|---|---|---|---|---|---|---|
| | | | | | | | High-Al spinel | | | | | | | | | | |
| $SiO_2$ | 0.10 | 0.03 | 0.08 | 0.05 | 0.12 | 0.06 | 0.03 | 0.06 | 0.04 | 0.04 | 0.10 | 0.10 | 0.04 | 0.02 | 0.07 | 0.04 | 0.04 |
| $TiO_2$ | 0.08 | 0.10 | 0.08 | 0.27 | 0.17 | 0.16 | 0.00 | 0.02 | 0.09 | 0.00 | 0.03 | 0.01 | 0.02 | 0.02 | 0.00 | 0.00 | 0.03 |
| $Al_2O_3$ | 24.93 | 25.29 | 25.18 | 25.00 | 25.11 | 24.81 | 23.88 | 24.10 | 24.23 | 23.21 | 27.92 | 30.58 | 24.42 | 24.10 | 22.47 | 21.74 | 21.83 |
| $Fe_2O_3$ | 1.50 | 1.39 | 1.58 | 1.56 | 1.94 | 2.07 | 1.18 | 0.84 | 0.71 | 0.92 | 3.09 | 3.03 | 1.86 | 1.87 | 2.26 | 2.55 | 2.17 |
| $Cr_2O_3$ | 42.16 | 42.79 | 43.06 | 43.30 | 43.43 | 43.39 | 44.38 | 45.00 | 44.65 | 45.17 | 37.47 | 34.99 | 43.87 | 43.74 | 46.00 | 46.13 | 46.50 |
| $V_2O_3$ | 0.21 | 0.25 | 0.29 | 0.12 | 0.06 | 0.18 | 0.20 | 0.19 | 0.20 | 0.19 | 0.34 | 0.33 | 0.26 | 0.24 | 0.20 | 0.17 | 0.17 |
| FeO | 17.49 | 17.42 | 17.44 | 14.72 | 14.84 | 14.68 | 18.32 | 18.05 | 17.94 | 18.04 | 18.22 | 17.67 | 17.19 | 17.35 | 16.71 | 16.76 | 17.02 |
| MgO | 11.49 | 11.94 | 12.05 | 13.70 | 13.81 | 13.81 | 11.22 | 11.39 | 11.51 | 11.15 | 11.43 | 12.00 | 11.93 | 11.75 | 12.19 | 11.97 | 11.94 |
| MnO | 0.01 | 0.02 | 0.01 | 0.00 | 0.00 | 0.00 | 0.00 | 0.04 | 0.00 | 0.00 | 0.03 | 0.07 | 0.00 | 0.00 | 0.05 | 0.02 | 0.00 |
| CoO | 0.08 | 0.17 | 0.06 | 0.09 | 0.12 | 0.03 | 0.11 | 0.14 | 0.15 | 0.09 | 0.08 | 0.15 | 0.11 | 0.12 | 0.10 | 0.10 | 0.11 |
| CuO | 0.14 | 0.00 | 0.08 | 0.04 | 0.00 | 0.00 | 0.00 | 0.09 | 0.00 | 0.00 | 0.00 | 0.04 | 0.09 | 0.11 | 0.04 | 0.02 | 0.00 | 0.00 |
| ZnO | 0.41 | 0.14 | 0.20 | 0.06 | 0.05 | 0.03 | 0.23 | 0.27 | 0.21 | 0.29 | 0.45 | 0.48 | 0.30 | 0.25 | 0.20 | 0.21 | 0.14 |
| NiO | 0.14 | 0.09 | 0.06 | 0.24 | 0.17 | 0.15 | 0.13 | 0.08 | 0.07 | 0.11 | 0.15 | 0.12 | 0.16 | 0.07 | 0.13 | 0.03 | 0.00 |
| Total | 98.74 | 99.63 | 100.17 | 99.15 | 99.82 | 99.37 | 99.68 | 100.27 | 99.80 | 99.21 | 99.35 | 99.62 | 100.27 | 99.57 | 100.40 | 99.72 | 99.95 |
| Cr# | 53 | 53 | 53 | 54 | 54 | 54 | 56 | 56 | 55 | 57 | 47 | 43 | 55 | 55 | 58 | 59 | 59 |
| Mg# | 54 | 53 | 50 | 62 | 62 | 63 | 52 | 53 | 53 | 52 | 53 | 55 | 55 | 55 | 57 | 56 | 56 |
| Cr/Fe | 2.12 | 2.19 | 2.17 | 2.54 | 2.47 | 2.48 | 2.18 | 2.26 | 2.27 | 2.26 | 1.69 | 1.62 | 2.20 | 2.17 | 2.32 | 2.29 | 2.32 |
| $Al_2O_{3\ melt}$ * | 15.58 | 15.64 | 15.62 | 15.60 | 15.61 | 15.56 | 15.41 | 15.45 | 15.47 | 15.29 | 16.05 | 16.43 | 15.50 | 15.44 | 15.15 | 15.02 | 15.04 |
| $TiO_{2\ melt}$ * | - | 0.01 | - | 0.73 | 0.40 | 0.34 | - | - | - | - | - | - | - | - | - | - | - |
| $FeO/MgO_{melt}$ * | 0.37 | 0.32 | 0.31 | 0.01 | 0.01 | - | 0.42 | 0.39 | 0.38 | 0.40 | 0.45 | 0.41 | 0.29 | 0.31 | 0.32 | 0.34 | 0.35 |
| | | | | | | | Formula, calculated on the basis of 32 oxygens (a.p.f.u.) | | | | | | | | | | |
| Si | 0.02 | 0.01 | 0.02 | 0.01 | 0.03 | 0.01 | 0.01 | 0.01 | 0.01 | 0.01 | 0.02 | 0.02 | 0.01 | 0.01 | 0.02 | 0.01 | 0.01 |
| Ti | 0.02 | 0.02 | 0.01 | 0.05 | 0.03 | 0.03 | 0.00 | 0.00 | 0.02 | 0.00 | 001 | 0.00 | 0.00 | 0.00 | 0.00 | 0.00 | 0.01 |
| Al | 7.31 | 7.33 | 7.26 | 7.20 | 7.18 | 7.13 | 6.96 | 7.00 | 7.05 | 6.84 | 8.06 | 8.68 | 7.07 | 7.03 | 6.53 | 6.38 | 6.40 |
| $Fe^{3+}$ | 0.28 | 0.26 | 0.29 | 0.29 | 0.36 | 0.38 | 0.22 | 0.15 | 0.13 | 0.17 | 0.57 | 0.55 | 0.34 | 0.35 | 0.42 | 0.48 | 0.41 |
| Cr | 8.29 | 8.32 | 8.33 | 8.37 | 8.33 | 8.37 | 8.76 | 8.77 | 8.72 | 8.93 | 7.25 | 6.66 | 8.52 | 8.56 | 8.97 | 9.09 | 9.14 |
| V | 0.04 | 0.05 | 0.06 | 0.02 | 0.01 | 0.04 | 0.04 | 0.04 | 0.04 | 0.04 | 0.07 | 0.06 | 0.05 | 0.05 | 0.04 | 0.03 | 0.03 |
| $Fe^{2+}$ | 3.64 | 3.58 | 3.57 | 3.01 | 3.01 | 2.99 | 3.79 | 3.72 | 3.71 | 3.77 | 3.73 | 3.56 | 3.53 | 3.59 | 3.45 | 3.49 | 3.54 |
| Mg | 4.26 | 4.38 | 4.39 | 4.99 | 4.99 | 5.02 | 4.14 | 4.19 | 4.24 | 4.15 | 4.17 | 4.31 | 4.37 | 4.33 | 4.48 | 4.45 | 4.42 |
| Mn | 0.00 | 0.00 | 0.00 | 0.00 | 0.00 | 0.00 | 0.00 | 0.01 | 0.00 | 0.00 | 0.01 | 0.01 | 0.00 | 0.00 | 0.01 | 0.00 | 0.00 |
| Co | 0.02 | 0.03 | 0.01 | 0.02 | 0.02 | 0.01 | 0.02 | 0.03 | 0.03 | 0.02 | 0.02 | 0.03 | 0.02 | 0.02 | 0.02 | 0.02 | 0.02 |
| Cu | 0.03 | 0.00 | 0.01 | 0.01 | 0.00 | 0.00 | 0.00 | 0.02 | 0.00 | 0.00 | 0.00 | 0.01 | 0.02 | 0.02 | 0.01 | 0.00 | 0.00 |
| Zn | 0.07 | 0.03 | 0.04 | 0.01 | 0.01 | 0.01 | 0.04 | 0.05 | 0.04 | 0.05 | 0.08 | 0.09 | 0.05 | 0.05 | 0.04 | 0.04 | 0.03 |
| Ni | 0.01 | 0.01 | 0.01 | 0.02 | 0.02 | 0.01 | 0.01 | 0.01 | 0.01 | 0.01 | 0.01 | 0.01 | 0.01 | 0.01 | 0.01 | 0.00 | 0.00 |

* Note: See text for calculation methods (chapter 5.1). Parental melts and tectonic settings. a.p.f.u—atom per formula unit.

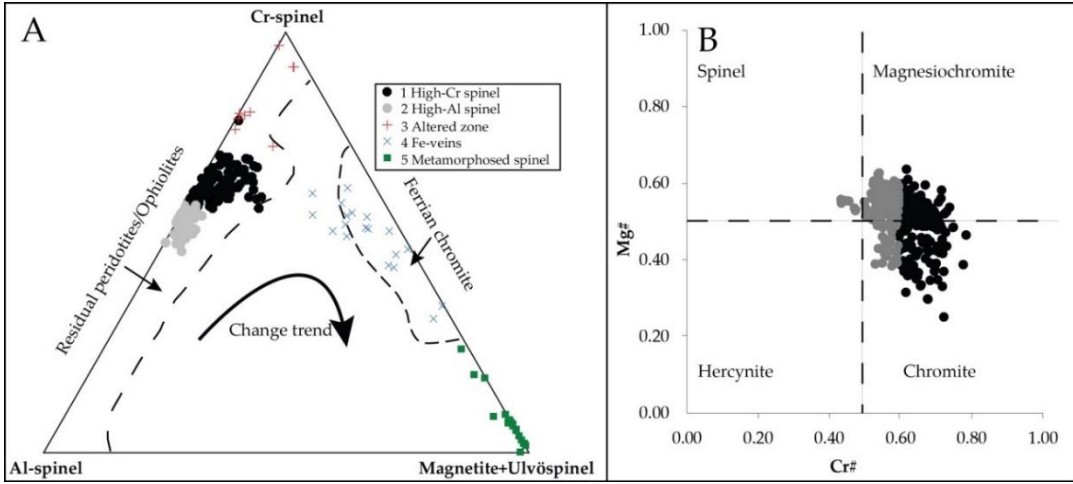

**Figure 6.** (**A**) The Cr-Al-Fe$^{3+}$ ternary diagram modified from [32] with distribution of spinel samples results; (**B**) Cr# vs. Mg# plot modified from [33] with distribution of the analyzed chromian spinels results. Black dots: high-Cr spinels, grey dots: high-Al spinels.

The high-Cr spinels (Table 3) show a chemically wide range of Cr$_2$O$_3$ content (42.03–52.63 wt%) with the highest Cr# (60–70). Major oxides (wt%) are present in varied amounts: Al$_2$O$_3$ (14.87–21.42), MgO (7.23–11.93), Fe$_2$O$_3$ (0.03–9.38), FeO (17.05–23.06), MnO (0.04–0.15). Low content of TiO$_2$ (0.30 wt%) is typical for podiform chromitites (Figure 7A,B). A small positive Pearson's correlation coefficient has been observed between Cr$_2$O$_3$ vs. TiO$_2$ (R = 0.19) and a medium positive correlation between Cr$_2$O$_3$ vs. Al$_2$O$_3$ (R = 0.44). The Mg# index (varying between 36–56) was calculated from the formula Mg# = [100·Mg/(Mg + Fe$^{2+}$)]. Minor amounts of V$_2$O$_3$ (0.11–0.26 wt%), ZnO (0.03–0.58 wt%), CoO (0.03–0.16 wt%), NiO (0.04–0.14 wt%) and CuO (0.01–0.10 wt%) were detected.

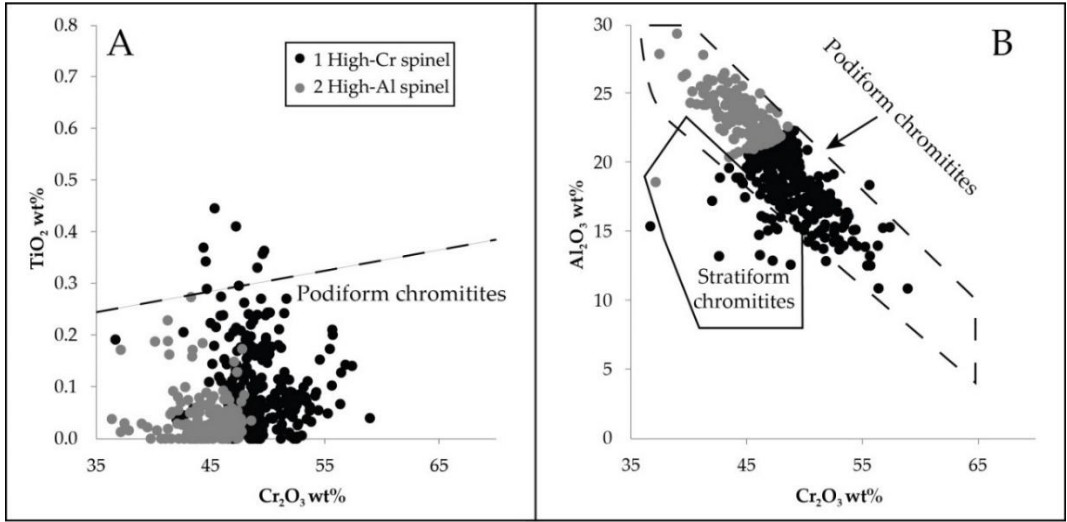

**Figure 7.** Chemical variations of investigated chromian spinels. (**A**) Cr$_2$O$_3$ vs. TiO$_2$; (**B**) Cr$_2$O$_3$ vs. Al$_2$O$_3$. Classification fields after [31]. Black dots: high-Cr spinels, grey dots: high-Al spinels.

The high-Al spinels show a wide range of chemical composition (Table 3): Cr$_2$O$_3$ (34.99–46.50 wt%), with Cr# 43–59; MgO (11.15–13.81 wt%), with Mg# 50–63, FeO (14.68–18.32 wt%), Fe$_2$O$_3$ (0.71–3.09 wt%), MnO (0.01–0.07 wt%), V$_2$O$_3$ (0.06–0.34 wt%). Transition metals in spinels are present in lower amounts: CoO (0.03–0.17 wt%), NiO (0.03–0.24 wt%), ZnO (0.03–0.48 wt%) and CuO (0.02–0.14 wt%). Microscopic mapping (SEM-BSE) reveals single phases of PGMs (Figure 8) in spinels as Cu-Os phases incorporated in Fe-veins (up to 2 µm). These phases were not analyzed by the EPMA method.

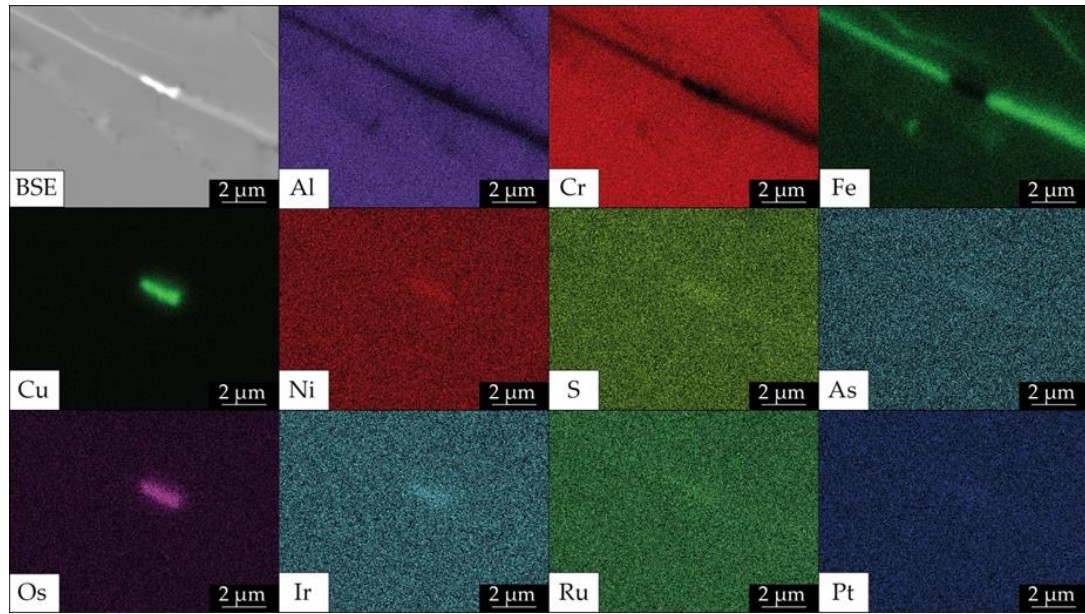

**Figure 8.** SEM-BSE showing the distribution of Al, Cr, Fe, Cu, Ni, S, As, Os, Ir, Ru, Ptin chromian spinels. The Fe-veins include the Cu-Os phases.

### 4.3. Altered Chromian Spinels

Chromian spinels, contrary to other minerals, are relatively resistant to hydrothermal fluids and atmospheric conditions. Their high resistance to external factors implies the ability to preserve original mineral properties [34–36]. Chromian spinels were transformed during post-magmatic evolution either in a low-grade metamorphic process [37] or through retrograde metamorphism [38,39].

The identified ferrian chromite is the result of post-magmatic metamorphic transformation. Some chromites have a visible zonal structure: ferrian chromite rim and Cr-magnetite rim (often skeletal Cr-magnetite). Many chromites have small, internal magnetite veins. Ferrian chromite grains (Table 4) have high content (in wt%) of FeO (19.42–29.82), $Fe_2O_3$ (18.02–51.27) and MnO (0.29–8.42) and low content of $Al_2O_3$ (0.27–7.81) and MgO (1.12–4.25). The $TiO_2$ content is below 0.54 wt%. Concentrations of transition metals are higher (ZnO 0.18–1.70 wt%, and NiO 0.10–0.57 wt%) than in unaltered spinels. This suggests that these elements were mobile during alteration. The chemical composition of veins is characterized by $Fe^{3+}$ and Cr-enrichments.

**Table 4.** The chemical composition of Cr magnetite, ferrian chromite and metamorphic altered chromian spinels (in wt %) with EPMA.

| | | | | | | **Cr-Magnetite Rim and Veins** | | | | | | | |
|---|---|---|---|---|---|---|---|---|---|---|---|---|---|
| **Sample** | **JP09-43** | **JP09-44** | **JP09-55** | **JP09-62** | **JP09-74** | **JP09-88** | **JP09-89** | **JP10-12** | **JP11-13** | **JP13-01** | **JP13-15** | **JP13-21** | **JP13-25** |
| $SiO_2$ | 0.08 | 0.13 | 0.05 | 0.16 | 0.11 | 0.35 | 0.94 | 0.47 | 0.64 | 0.45 | 0.09 | 1.28 | 0.05 |
| $TiO_2$ | 0.01 | 0.03 | 0.02 | 0.00 | 0.00 | 0.00 | 0.06 | 0.02 | 0.00 | 0.01 | 0.04 | 0.04 | 0.85 |
| $Al_2O_3$ | 0.00 | 0.00 | 0.00 | 0.09 | 0.06 | 0.04 | 0.27 | 0.09 | 0.24 | 0.07 | 0.15 | 0.00 | 4.19 |
| $Fe_2O_3$ | 65.33 | 67.82 | 67.91 | 63.66 | 65.66 | 65.54 | 60.70 | 64.68 | 65.21 | 66.89 | 56.48 | 64.60 | 44.10 |
| $Cr_2O_3$ | 3.64 | 1.06 | 1.25 | 5.14 | 3.48 | 2.45 | 4.41 | 2.64 | 1.59 | 1.68 | 11.55 | 1.32 | 17.72 |
| $V_2O_3$ | 0.05 | 0.04 | 0.03 | 0.01 | 0.03 | 0.03 | 0.03 | 0.07 | 0.06 | 0.00 | 0.05 | 0.06 | 0.27 |
| FeO | 30.47 | 30.95 | 30.50 | 30.54 | 30.75 | 30.35 | 31.23 | 30.08 | 29.77 | 31.04 | 29.34 | 31.90 | 27.66 |
| MgO | 0.27 | 0.15 | 0.30 | 0.33 | 0.20 | 0.16 | 0.17 | 0.52 | 0.93 | 0.30 | 0.40 | 0.41 | 2.69 |
| MnO | 0.11 | 0.07 | 0.20 | 0.09 | 0.11 | 0.12 | 0.15 | 0.04 | 0.14 | 0.02 | 1.02 | 0.10 | 0.19 |
| CoO | 0.04 | 0.02 | 0.02 | 0.14 | 0.12 | 0.10 | 0.14 | 0.21 | 0.15 | 0.09 | 0.06 | 0.01 | 0.01 |
| CuO | 0.02 | 0.01 | 0.00 | 0.11 | 0.10 | 0.12 | 0.00 | 0.09 | 0.00 | 0.10 | 0.00 | 0.00 | 0.00 |
| ZnO | 0.03 | 0.00 | 0.05 | 0.00 | 0.00 | 0.03 | 0.05 | 0.14 | 0.00 | 0.13 | 0.04 | 0.00 | 0.03 |
| NiO | 0.47 | 0.18 | 0.07 | 0.10 | 0.11 | 0.05 | 0.04 | 0.15 | 0.14 | 0.11 | 0.23 | 0.05 | 0.64 |
| Total | 100.52 | 100.46 | 100.40 | 100.37 | 100.73 | 98.34 | 98.19 | 99.20 | 98.87 | 100.89 | 99.45 | 99.77 | 98.40 |
| | | | | | Formula, calculated on the basis of 32 oxygens (a.p.f.u.) | | | | | | | | |
| Si | 0.02 | 0.04 | 0.01 | 0.05 | 0.03 | 0.11 | 0.29 | 0.14 | 0.20 | 0.14 | 0.03 | 0.39 | 0.01 |
| Ti | 0.00 | 0.01 | 0.00 | 0.00 | 0.00 | 0.00 | 0.01 | 0.01 | 0.00 | 0.00 | 0.01 | 0.01 | 0.19 |
| Al | 0.00 | 0.00 | 0.00 | 0.03 | 0.02 | 0.01 | 0.10 | 0.03 | 0.09 | 0.02 | 0.05 | 0.00 | 1.47 |
| $Fe^{3+}$ | 15.05 | 15.64 | 15.65 | 14.63 | 15.06 | 15.16 | 14.20 | 15.01 | 15.11 | 15.29 | 13.05 | 14.87 | 9.88 |
| Cr | 0.88 | 0.26 | 0.30 | 1.24 | 0.84 | 0.60 | 1.08 | 0.64 | 0.39 | 0.40 | 2.80 | 0.32 | 4.17 |
| V | 0.01 | 0.01 | 0.01 | 0.00 | 0.01 | 0.01 | 0.01 | 0.02 | 0.02 | 0.00 | 0.01 | 0.01 | 0.06 |
| $Fe^{2+}$ | 7.80 | 7.93 | 7.81 | 7.80 | 7.84 | 7.92 | 8.12 | 7.76 | 7.67 | 7.89 | 7.54 | 8.16 | 6.88 |
| Mg | 0.12 | 0.07 | 0.13 | 0.15 | 0.09 | 0.08 | 0.08 | 0.24 | 0.43 | 0.14 | 0.18 | 0.19 | 1.20 |
| Mn | 0.03 | 0.02 | 0.05 | 0.02 | 0.03 | 0.03 | 0.04 | 0.01 | 0.04 | 0.01 | 0.26 | 0.03 | 0.05 |
| Co | 0.01 | 0.00 | 0.00 | 0.03 | 0.03 | 0.02 | 0.04 | 0.05 | 0.04 | 0.02 | 0.02 | 0.00 | 0.00 |
| Cu | 0.00 | 0.00 | 0.00 | 0.02 | 0.02 | 0.03 | 0.00 | 0.02 | 0.00 | 0.02 | 0.00 | 0.00 | 0.00 |
| Zn | 0.01 | 0.00 | 0.01 | 0.00 | 0.00 | 0.01 | 0.01 | 0.03 | 0.00 | 0.03 | 0.01 | 0.00 | 0.01 |
| Ni | 0.05 | 0.02 | 0.01 | 0.01 | 0.01 | 0.01 | 0.00 | 0.02 | 0.02 | 0.01 | 0.03 | 0.01 | 0.07 |

**Table 4.** *Cont.*

| | | | | | Ferrian chromite | | | | | | | | |
|---|---|---|---|---|---|---|---|---|---|---|---|---|---|
| **Sample** | **JP09-43** | **JP09-45** | **JP09-57** | **JP09-67** | **JP09-76** | **JP09-88** | **JP10-14** | **JP11-17** | **JP13-10** | **JP13-17** | **JP13-26** | **JP13-28** | **JP49-16** |
| $SiO_2$ | 0.12 | 0.05 | 0.03 | 0.04 | 0.03 | 0.05 | 0.13 | 0.11 | 0.05 | 0.03 | 0.04 | 0.44 | 0.13 |
| $TiO_2$ | 0.18 | 0.35 | 0.54 | 0.26 | 0.22 | 0.27 | 0.00 | 0.29 | 0.12 | 0.54 | 0.26 | 0.04 | 0.03 |
| $Al_2O_3$ | 2.69 | 3.20 | 4.26 | 7.81 | 3.03 | 6.54 | 0.73 | 5.51 | 2.48 | 4.26 | 7.81 | 0.35 | 0.27 |
| $Fe_2O_3$ | 36.28 | 28.18 | 24.15 | 20.08 | 34.89 | 18.02 | 51.27 | 26.43 | 27.28 | 24.32 | 20.20 | 35.21 | 44.62 |
| $Cr_2O_3$ | 28.95 | 36.21 | 37.68 | 39.67 | 30.26 | 42.89 | 16.22 | 35.17 | 37.55 | 37.68 | 39.67 | 31.79 | 22.85 |
| $V_2O_3$ | 0.42 | 0.45 | 0.47 | 0.49 | 0.13 | 0.18 | 0.03 | 0.49 | 0.55 | 0.47 | 0.49 | 0.08 | 0.05 |
| FeO | 29.82 | 28.35 | 27.53 | 25.77 | 28.91 | 27.77 | 27.59 | 26.71 | 28.01 | 27.37 | 25.66 | 19.42 | 23.19 |
| MgO | 1.15 | 2.18 | 2.66 | 4.25 | 1.67 | 2.92 | 1.12 | 3.30 | 1.92 | 2.66 | 4.25 | 1.31 | 1.12 |
| MnO | 0.47 | 0.36 | 0.30 | 0.29 | 0.42 | 0.31 | 1.06 | 0.35 | 0.51 | 0.30 | 0.29 | 8.42 | 5.29 |
| CoO | 0.09 | 0.06 | 0.05 | 0.06 | 0.04 | 0.09 | 0.20 | 0.10 | 0.07 | 0.05 | 0.06 | 0.18 | 0.23 |
| CuO | 0.00 | 0.00 | 0.02 | 0.04 | 0.00 | 0.00 | 0.13 | 0.00 | 0.00 | 0.02 | 0.04 | 0.06 | 0.03 |
| ZnO | 0.18 | 0.22 | 0.20 | 0.31 | 0.00 | 0.31 | 0.35 | 0.22 | 0.28 | 0.20 | 0.31 | 1.70 | 0.67 |
| NiO | 0.21 | 0.36 | 0.31 | 0.21 | 0.22 | 0.11 | 0.57 | 0.34 | 0.25 | 0.31 | 0.21 | 0.10 | 0.11 |
| Total | 100.56 | 99.97 | 98.20 | 99.28 | 99.82 | 99.46 | 99.40 | 99.02 | 99.07 | 98.21 | 99.29 | 99.10 | 98.59 |
| Cr# | 88 | 88 | 86 | 77 | 87 | 81 | 94 | 81 | 91 | 86 | 77 | 98 | 98 |
| Mg# | 6 | 12 | 15 | 23 | 9 | 16 | 7 | 18 | 11 | 9 | 15 | 11 | 8 |
| Cr/Fe | 0.44 | 0.64 | 0.72 | 0.86 | 0.47 | 0.92 | 0.21 | 0.66 | 0.68 | 0.72 | 0.86 | 0.59 | 0.34 |
| | | | | | Formula, calculated on the basis of 32 oxygens (a.p.f.u.) | | | | | | | | |
| Si | 0.03 | 0.01 | 0.01 | 0.01 | 0.01 | 0.01 | 0.04 | 0.03 | 0.02 | 0.01 | 0.01 | 0.13 | 0.04 |
| Ti | 0.04 | 0.08 | 0.12 | 0.06 | 0.05 | 0.06 | 0.00 | 0.06 | 0.03 | 0.12 | 0.06 | 0.01 | 0.01 |
| Al | 0.93 | 1.10 | 1.48 | 2.60 | 1.05 | 2.21 | 0.26 | 1.88 | 0.87 | 1.48 | 2.60 | 0.12 | 0.10 |
| $Fe^{3+}$ | 8.06 | 6.21 | 5.36 | 4.27 | 7.74 | 3.88 | 11.75 | 5.76 | 6.10 | 5.39 | 4.29 | 7.99 | 10.27 |
| Cr | 6.76 | 8.39 | 8.79 | 8.87 | 7.05 | 9.71 | 3.91 | 8.06 | 8.82 | 8.77 | 8.86 | 7.58 | 5.53 |
| V | 0.10 | 0.11 | 0.11 | 0.11 | 0.03 | 0.04 | 0.01 | 0.11 | 0.13 | 0.11 | 0.11 | 0.02 | 0.01 |
| $Fe^{2+}$ | 7.36 | 6.95 | 6.79 | 6.10 | 7.13 | 6.65 | 7.03 | 6.47 | 6.96 | 6.74 | 6.06 | 4.90 | 5.93 |
| Mg | 0.51 | 0.95 | 1.17 | 1.79 | 0.74 | 1.25 | 0.51 | 1.42 | 0.85 | 1.17 | 1.79 | 0.64 | 0.51 |
| Mn | 0.12 | 0.09 | 0.08 | 0.07 | 0.10 | 0.08 | 0.27 | 0.09 | 0.13 | 0.08 | 0.07 | 2.15 | 1.37 |
| Co | 0.02 | 0.01 | 0.01 | 0.01 | 0.01 | 0.02 | 0.05 | 0.02 | 0.02 | 0.01 | 0.01 | 0.04 | 0.06 |
| Cu | 0.00 | 0.00 | 0.00 | 0.01 | 0.00 | 0.00 | 0.03 | 0.00 | 0.00 | 0.00 | 0.01 | 0.01 | 0.01 |
| Zn | 0.04 | 0.05 | 0.04 | 0.06 | 0.03 | 0.06 | 0.08 | 0.05 | 0.06 | 0.04 | 0.06 | 0.38 | 0.15 |
| Ni | 0.02 | 0.04 | 0.03 | 0.02 | 0.05 | 0.01 | 0.06 | 0.04 | 0.03 | 0.07 | 0.05 | 0.01 | 0.01 |

**Table 4.** *Cont.*

| | Metamorphic altered spinels | | | | | | | | | | | | |
|---|---|---|---|---|---|---|---|---|---|---|---|---|---|
| Sample | JP09-11 | JP09-12 | JP09-13 | JP07-14 | JP07-17 | JP07-24 | JP07-32 | JP07-34 | JP07-32 | JP07-34 | JP07-36 | JP13-1 | JP13-2 |
| $SiO_2$ | 0.35 | 0.66 | 0.90 | 0.05 | 0.00 | 0.01 | 0.08 | 0.01 | 0.00 | 0.00 | 0.09 | 0.09 | 0.05 |
| $TiO_2$ | 0.20 | 0.12 | 0.07 | 0.02 | 0.07 | 0.05 | 0.01 | 0.12 | 0.07 | 0.05 | 0.08 | 0.07 | 0.02 |
| $Al_2O_3$ | 1.26 | 1.14 | 10.23 | 10.53 | 8.53 | 11.18 | 1.56 | 0.58 | 8.53 | 9.41 | 9.20 | 10.23 | 10.53 |
| $Fe_2O_3$ | 4.07 | 4.09 | 0.00 | 0.25 | 1.57 | 0.88 | 0.00 | 0.58 | 1.57 | 0.96 | 0.00 | 0.00 | 0.25 |
| $Cr_2O_3$ | 62.81 | 61.76 | 63.26 | 61.36 | 60.46 | 58.07 | 69.33 | 58.64 | 60.46 | 60.60 | 63.75 | 64.26 | 61.36 |
| $V_2O_3$ | 0.17 | 0.22 | 0.00 | 0.00 | 0.05 | 0.22 | 0.06 | 0.19 | 0.05 | 0.09 | 0.13 | 0.00 | 0.00 |
| FeO | 28.38 | 28.51 | 13.03 | 13.87 | 19.31 | 18.56 | 21.22 | 20.32 | 19.31 | 17.81 | 13.33 | 13.03 | 13.87 |
| MgO | 2.61 | 2.52 | 11.10 | 12.56 | 8.72 | 9.52 | 6.74 | 8.13 | 8.72 | 9.84 | 11.68 | 11.10 | 12.56 |
| MnO | 0.28 | 0.29 | 0.25 | 0.23 | 0.35 | 0.30 | 0.00 | 0.43 | 0.00 | 0.03 | 0.01 | 0.25 | 0.23 |
| CoO | 0.06 | 0.11 | 0.00 | 0.00 | 0.00 | 0.00 | 0.08 | 0.05 | 0.00 | 0.00 | 0.10 | 0.00 | 0.00 |
| CuO | 0.08 | 0.06 | 0.00 | 0.00 | 0.04 | 0.04 | 0.00 | 0.00 | 0.04 | 0.00 | 0.00 | 0.00 | 0.00 |
| ZnO | 0.20 | 0.23 | 0.00 | 0.21 | 0.23 | 0.09 | 0.18 | 0.21 | 0.23 | 0.14 | 0.20 | 0.10 | 0.21 |
| NiO | 0.11 | 0.11 | 0.00 | 0.00 | 0.00 | 0.00 | 0.02 | 0.03 | 0.00 | 0.00 | 0.07 | 0.00 | 0.00 |
| Total | 100.58 | 99.83 | 98.14 | 99.08 | 99.33 | 98.91 | 99.29 | 98.91 | 99.33 | 99.22 | 98.64 | 99.13 | 99.08 |
| Cr# | 97 | 97 | 81 | 80 | 83 | 78 | 97 | 79 | 83 | 81 | 82 | 81 | 80 |
| Mg# | 14 | 14 | 60 | 62 | 45 | 48 | 36 | 42 | 45 | 50 | 61 | 60 | 61 |
| Cr/Fe | 1.85 | 1.81 | 4.59 | 4.11 | 2.76 | 2.84 | 3.09 | 2.66 | 2.76 | 3.07 | 4.52 | 4.66 | 4.11 |
| Formula, calculated on the basis of 32 oxygens (a.p.f.u.) | | | | | | | | | | | | | |
| Si | 0.10 | 0.19 | 0.02 | 0.01 | 0.00 | 0.00 | 0.02 | 0.00 | 0.00 | 0.00 | 0.02 | 0.02 | 0.01 |
| Ti | 0.04 | 0.03 | 0.01 | 0.00 | 0.01 | 0.01 | 0.00 | 0.02 | 0.01 | 0.01 | 0.02 | 0.01 | 0.00 |
| Al | 0.43 | 0.39 | 3.22 | 3.24 | 2.72 | 3.51 | 0.52 | 3.25 | 2.72 | 2.96 | 2.88 | 3.19 | 3.24 |
| $Fe^{3+}$ | 0.99 | 0.99 | 0.00 | 0.05 | 0.32 | 0.18 | 0.00 | 0.12 | 0.32 | 0.19 | 0.00 | 0.00 | 0.05 |
| Cr | 14.36 | 14.22 | 13.34 | 12.67 | 12.92 | 12.24 | 15.51 | 12.54 | 12.92 | 12.81 | 13.39 | 13.43 | 2.67 |
| V | 0.04 | 0.05 | 0.00 | 0.00 | 0.01 | 0.05 | 0.01 | 0.04 | 0.01 | 0.02 | 0.03 | 0.00 | 0.00 |
| $Fe^{2+}$ | 6.86 | 6.95 | 2.91 | 3.03 | 4.37 | 4.14 | 5.02 | 4.59 | 4.37 | 3.98 | 2.96 | 2.88 | 3.03 |
| Mg | 1.12 | 1.09 | 4.41 | 4.89 | 3.52 | 3.78 | 2.84 | 3.28 | 3.51 | 3.92 | 4.63 | 4.38 | 4.89 |
| Mn | 0.07 | 0.07 | 0.06 | 0.05 | 0.08 | 0.07 | 0.00 | 0.10 | 0.08 | 0.07 | 0.00 | 0.06 | 0.05 |
| Co | 0.01 | 0.03 | 0.00 | 0.00 | 0.00 | 0.00 | 0.02 | 0.01 | 0.00 | 0.00 | 0.02 | 0.00 | 0.00 |
| Cu | 0.02 | 0.01 | 0.00 | 0.00 | 0.01 | 0.01 | 0.00 | 0.00 | 0.01 | 0.00 | 0.00 | 0.00 | 0.00 |
| Zn | 0.04 | 0.05 | 0.02 | 0.04 | 0.05 | 0.02 | 0.04 | 0.04 | 0.05 | 0.03 | 0.04 | 0.02 | 0.00 |
| Ni | 0.01 | 0.01 | 0.00 | 0.00 | 0.00 | 0.00 | 0.01 | 0.00 | 0.00 | 0.00 | 0.01 | 0.00 | 0.00 |

The changes recognized in chromites probably occurred during alteration of the Cyclops ophiolite in a retrograde evolution (most likely in the amphibolite facies). The microstructure and chemical composition of metamorphosed chromites are different from other unaltered chromites (Table 4). The grains display internal fractures. Metamorphic grains (Table 4) of Cr-spinels have high values of $Cr_2O_3$ (58.07–69.33 wt%), FeO (13.03–28.51 wt%) and Cr# (78–97) and low content of $Al_2O_3$ (0.58–11.18 wt%), MgO (2.52–12.56 wt%), $V_2O_3$ (0.05–0.22 wt%) and MnO (0.01–0.43 wt%). The ranges of minor metals are as follows: ZnO (0.09–0.23 wt%), CoO (0.05–0.11 wt%), and NiO (0.02–0.11 wt%).

### 4.4. Olivine

Olivine was identified in two forms: (a) unaltered olivine grains and (b) serpentinized grains. Both grains are present in all grain classes (0.50–0.063 mm). Olivine coexists in paragenesis with chromite, magnetite and Cr-garnet (Figure 3). Unaltered olivine is composed of forsterite (Fo 90–93) and contains NiO (0.32–0.55 wt%) (Table 5). CaO content is low (0.01–0.04 wt%). In olivine, the magnesium value Mg# is 90–93 and is close to olivine from mid-ocean ridge peridotites. A similar Mg# value is presented by [5]. Olivine originates from dunites and harzburgites of the Cyclops Mountains. They present high Mg# (90–92) and high content of NiO (0.38–0.45 wt%) [5]. The geochemical composition of olivine from the Cyclops Mountains massif indicates a similar source.

Olivine is partly replaced by pseudomorphic mesh textures. Brownish veins of serpentine-group minerals crosscut the minerals. Serpentinization was not completed. The effect of changes was the formation of serpentine group minerals: lizardite and antigorite (identified by XRD analysis). The development of olivine serpentinization was accompanied by growth of magnetite ($Fe^{2+}Fe^{3+}_2O_4$) and singular phases of very fine awaruite ($Ni_3Fe$) and heazlewoodite ($Ni_3S_2$). Fine-grained opaque minerals are observed along the margins and interstitial cracks system of individual mesh cells. Minerals have been identified during SEM observations. Ingrowths of awaruite and heazlewoodite create small anhedral forms. The size of the awaruite is up to 10 μm, heazlewoodite is up to 3 μm, and the size of magnetite is up to 0.10 mm. Serpentine-group minerals contain high amounts of NiO (0.02–1.43) and $Cr_2O_3$ (0.02–1.83) (in wt%) (Table 5).

### 4.5. Chromian Andradite

SEM observations can identify the presence of Cr-garnet in paragenesis with altered olivine and magnetite (Figure 3). Garnet is represented by anhedral crystals, rarely in subhedral forms. Garnet (Table 6) shows characteristic variability in $Cr_2O_3$ content. There is a visible negative correlation (R = −0.98) between $Fe_2O_3$ and $Cr_2O_3$. The concentration of $Cr_2O_3$ ranges from 0.02 to 9.71 wt% and $Fe_2O_3$ from 21.00 to 31.47 wt%. SEM observations reveal no compositional zonation in the grains.

A spreadsheet introduced by [40] was used to calculate the proportion of moles in garnets and indicate their end members. The calculation result shows dominance of andradite over uvarovite and schorlomite.

**Table 5.** Chemical composition of unaltered and altered olivine (in wt%) from EPMA analysis.

| | | | | | | | Unaltered Olivine | | | | | | |
|---|---|---|---|---|---|---|---|---|---|---|---|---|---|
| Sample | JP04-55 | JP06-53 | JP06-55 | JP06-87 | JP06-125 | JP07-69 | JP07-72 | JP07-78 | JP08-13 | JP08-21 | JP08-24 | JP08-30 | JP10-65 |
| $SiO_2$ | 41.49 | 41.12 | 41.25 | 41.53 | 41.55 | 41.71 | 42.21 | 41.00 | 41.32 | 41.27 | 41.13 | 40.87 | 41.29 |
| $TiO_2$ | 0.00 | 0.01 | 0.00 | 0.00 | 0.00 | 0.01 | 0.00 | 0.00 | 0.00 | 0.01 | 0.00 | 0.00 | 0.00 |
| $Al_2O_3$ | 0.02 | 0.00 | 0.01 | 0.01 | 0.00 | 0.00 | 0.08 | 0.00 | 0.00 | 0.00 | 0.00 | 0.00 | 0.00 |
| $Cr_2O_3$ | 0.01 | 0.10 | 0.03 | 0.01 | 0.06 | 0.03 | 0.05 | 0.00 | 0.00 | 0.04 | 0.21 | 0.20 | 0.05 |
| FeO | 8.23 | 8.33 | 8.04 | 7.94 | 7.83 | 7.23 | 8.99 | 8.44 | 7.91 | 8.03 | 6.73 | 8.00 | 8.46 |
| MgO | 50.18 | 49.70 | 49.91 | 49.94 | 50.42 | 49.99 | 47.33 | 49.18 | 49.48 | 50.24 | 50.74 | 50.38 | 49.25 |
| MnO | 0.13 | 0.10 | 0.05 | 0.11 | 0.20 | 0.13 | 0.21 | 0.03 | 0.14 | 0.13 | 0.16 | 0.09 | 0.17 |
| CaO | 0.00 | 0.01 | 0.01 | 0.00 | 0.04 | 0.00 | 0.01 | 0.02 | 0.03 | 0.01 | 0.01 | 0.01 | 0.01 |
| NiO | 0.50 | 0.36 | 0.38 | 0.32 | 0.42 | 0.38 | 0.34 | 0.39 | 0.38 | 0.55 | 0.49 | 0.42 | 0.55 |
| ZnO | 0.00 | 0.01 | 0.02 | 0.02 | 0.00 | 0.00 | 0.00 | 0.09 | 0.04 | 0.02 | 0.04 | 0.09 | 0.08 |
| Total | 100.55 | 99.74 | 99.69 | 99.89 | 100.48 | 99.48 | 99.22 | 99.14 | 99.30 | 100.31 | 99.51 | 100.05 | 99.86 |
| Fo | 91.45 | 91.32 | 91.67 | 91.71 | 91.80 | 92.37 | 90.17 | 91.19 | 91.64 | 91.65 | 92.92 | 91.74 | 91.05 |
| Fa | 8.41 | 8.58 | 8.28 | 8.18 | 8.00 | 7.49 | 9.60 | 8.77 | 8.22 | 8.22 | 6.91 | 8.17 | 8.78 |
| Mg# | 91 | 91 | 92 | 92 | 92 | 92 | 90 | 91 | 92 | 92 | 93 | 92 | 91 |
| | | | | | Formula, calculated on the basis of 4 oxygens (a.p.f.u.) | | | | | | | | |
| Si | 1.00 | 1.00 | 1.01 | 1.00 | 1.00 | 1.01 | 1.03 | 1.01 | 1.01 | 1.00 | 1.00 | 1.00 | 1.01 |
| Ti | 0.00 | 0.00 | 0.00 | 0.00 | 0.00 | 0.00 | 0.00 | 0.00 | 0.00 | 0.00 | 0.00 | 0.00 | 0.00 |
| Al | 0.00 | 0.00 | 0.00 | 0.00 | 0.00 | 0.00 | 0.00 | 0.00 | 0.00 | 0.00 | 0.00 | 0.00 | 0.00 |
| Cr | 0.00 | 0.00 | 0.00 | 0.00 | 0.00 | 0.00 | 0.00 | 0.00 | 0.00 | 0.00 | 0.00 | 0.00 | 0.00 |
| $Fe^{2+}$ | 0.17 | 0.17 | 0.16 | 0.17 | 0.17 | 0.15 | 0.18 | 0.17 | 0.16 | 0.16 | 0.14 | 0.16 | 0.17 |
| Mg | 1.81 | 1.81 | 1.81 | 1.81 | 1.81 | 1.81 | 1.73 | 1.80 | 1.81 | 1.82 | 1.84 | 1.83 | 1.79 |
| Mn | 0.00 | 0.00 | 0.00 | 0.00 | 0.00 | 0.00 | 0.00 | 0.00 | 0.00 | 0.00 | 0.00 | 0.00 | 0.00 |
| Ni | 0.01 | 0.01 | 0.01 | 0.01 | 0.01 | 0.01 | 0.01 | 0.01 | 0.01 | 0.01 | 0.01 | 0.01 | 0.01 |
| Ca | 0.00 | 0.00 | 0.00 | 0.00 | 0.00 | 0.00 | 0.00 | 0.00 | 0.00 | 0.00 | 0.00 | 0.00 | 0.00 |
| Zn | 0.00 | 0.00 | 0.00 | 0.00 | 0.00 | 0.00 | 0.00 | 0.00 | 0.00 | 0.00 | 0.00 | 0.00 | 0.00 |
| | | | | | Serpentine-group minerals | | | | | | | | |
| Sample | JP04-46 | JP04-50 | JP04-57 | JP04-58 | JP06-26 | JP06-44 | JP07-48 | JP07-52 | JP08-6 | JP08-7 | JP08-16 | JP082-35 | JP08-38 |
| $SiO_2$ | 42.39 | 43.59 | 43.02 | 43.72 | 43.91 | 42.87 | 38.62 | 46.55 | 38.06 | 40.92 | 35.30 | 42.73 | 44.11 |
| $TiO_2$ | 0.01 | 0.00 | 0.00 | 0.01 | 0.01 | 0.00 | 0.03 | 0.00 | 0.03 | 0.00 | 0.06 | 0.00 | 0.02 |
| $Al_2O_3$ | 0.30 | 0.14 | 0.25 | 0.33 | 0.45 | 0.56 | 0.23 | 0.26 | 0.60 | 0.41 | 0.53 | 0.48 | 0.27 |
| $Cr_2O_3$ | 0.07 | 0.02 | 0.02 | 0.16 | 0.27 | 0.23 | 0.12 | 0.10 | 1.83 | 0.16 | 1.19 | 0.19 | 0.11 |
| FeO | 5.09 | 2.19 | 3.70 | 1.78 | 2.35 | 8.06 | 2.39 | 3.26 | 3.44 | 3.22 | 2.63 | 4.46 | 1.43 |

**Table 5.** *Cont.*

| | | | | | | Unaltered Olivine | | | | | | | |
|---|---|---|---|---|---|---|---|---|---|---|---|---|---|
| **Sample** | **JP04-55** | **JP06-53** | **JP06-55** | **JP06-87** | **JP06-125** | **JP07-69** | **JP07-72** | **JP07-78** | **JP08-13** | **JP08-21** | **JP08-24** | **JP08-30** | **JP10-65** |
| MgO | 36.37 | 40.15 | 37.58 | 40.19 | 39.38 | 34.05 | 32.90 | 38.28 | 32.96 | 35.26 | 31.52 | 37.86 | 39.36 |
| MnO | 0.07 | 0.02 | 0.02 | 0.07 | 0.11 | 0.05 | 0.05 | 0.08 | 0.02 | 0.03 | 0.09 | 0.07 | 0.08 |
| CaO | 0.06 | 0.03 | 0.08 | 0.03 | 0.12 | 0.16 | 0.26 | 0.13 | 0.14 | 0.05 | 0.06 | 0.08 | 0.08 |
| NiO | 0.63 | 0.23 | 0.83 | 0.03 | 0.04 | 1.43 | 0.11 | 0.60 | 0.17 | 1.04 | 0.14 | 0.33 | 0.02 |
| ZnO | 0.05 | 0.00 | 0.00 | 0.00 | 0.02 | 0.00 | 0.06 | 0.00 | 0.06 | 0.02 | 0.08 | 0.00 | 0.01 |
| Total | 85.04 | 86.37 | 85.50 | 86.32 | 86.66 | 87.41 | 74.77 | 89.29 | 77.31 | 81.11 | 71.60 | 86.20 | 85.49 |
| Fo | 92.63 | 97.00 | 94.74 | 97.48 | 96.61 | 88.22 | 96.00 | 95.34 | 94.44 | 95.08 | 95.38 | 93.72 | 97.89 |
| Fa | 7.27 | 2.97 | 5.23 | 2.43 | 3.24 | 11.71 | 3.91 | 4.55 | 5.53 | 4.87 | 4.47 | 6.19 | 2.00 |
| Mg# | 93 | 97 | 95 | 97 | 97 | 88 | 96 | 95 | 94 | 95 | 95 | 94 | 98 |
| Formula, calculated on the basis of 7 oxygens (a.p.f.u.) | | | | | | | | | | | | | |
| Si | 2.053 | 2.043 | 2.057 | 2.044 | 2.051 | 2.057 | 2.090 | 2.114 | 2.022 | 2.062 | 2.016 | 2.033 | 2.074 |
| Ti | 0.000 | 0.000 | 0.000 | 0.000 | 0.000 | 0.000 | 0.001 | 0.000 | 0.001 | 0.000 | 0.003 | 0.000 | 0.001 |
| Al | 0.017 | 0.008 | 0.014 | 0.018 | 0.025 | 0.032 | 0.015 | 0.014 | 0.038 | 0.024 | 0.036 | 0.027 | 0.015 |
| Cr | 0.003 | 0.001 | 0.001 | 0.006 | 0.010 | 0.009 | 0.005 | 0.004 | 0.077 | 0.006 | 0.054 | 0.007 | 0.004 |
| $Fe^{2+}$ | 0.206 | 0.086 | 0.148 | 0.070 | 0.092 | 0.323 | 0.108 | 0.124 | 0.153 | 0.136 | 0.126 | 0.177 | 0.056 |
| Mg | 2.626 | 2.805 | 2.679 | 2.801 | 2.742 | 2.436 | 2.655 | 2.591 | 2.611 | 2.648 | 2.684 | 2.686 | 2.758 |
| Mn | 0.003 | 0.001 | 0.001 | 0.003 | 0.004 | 0.002 | 0.002 | 0.003 | 0.001 | 0.001 | 0.004 | 0.003 | 0.003 |
| Ni | 0.025 | 0.009 | 0.032 | 0.001 | 0.002 | 0.055 | 0.005 | 0.022 | 0.007 | 0.042 | 0.006 | 0.013 | 0.001 |
| Ca | 0.003 | 0.002 | 0.004 | 0.002 | 0.006 | 0.008 | 0.015 | 0.006 | 0.008 | 0.003 | 0.004 | 0.004 | 0.004 |
| Zn | 0.002 | 0.000 | 0.000 | 0.000 | 0.001 | 0.000 | 0.002 | 0.000 | 0.002 | 0.001 | 0.003 | 0.000 | 0.000 |
| Total | 4.937 | 4.953 | 4.936 | 4.944 | 4.932 | 4.923 | 4.899 | 4.878 | 4.920 | 4.923 | 4.936 | 4.950 | 4.916 |

Table 6. Chemical composition of chromian andradite (in wt%) from EPMA. Values of end members were calculated on the basis of a spreadsheet after [40].

| Sample | $SiO_2$ | $TiO_2$ | $Al_2O_3$ | $Cr_2O_3$ | $Fe_2O_3$ (cal) | $Mn_2O_3$ (cal) | MgO | CaO | $Na_2O$ | Total | End Members | | | |
| --- | --- | --- | --- | --- | --- | --- | --- | --- | --- | --- | --- | --- | --- | --- |
| | | | | | | | | | | | Schorlomite | Uvarovite | Andradite | Remainder |
| JP06-1 | 34.81 | 0.26 | 0.03 | 6.63 | 23.25 | 0.06 | 0.18 | 33.81 | 0.00 | 99.03 | 0.84 | 22.23 | 71.98 | 4.96 |
| JP06-2 | 35.12 | 0.16 | 0.10 | 5.39 | 24.73 | 0.06 | 0.17 | 33.83 | 0.02 | 99.56 | 0.49 | 17.97 | 77.21 | 4.33 |
| JP06-3 | 35.02 | 0.11 | 0.04 | 1.08 | 29.46 | 0.00 | 0.27 | 33.57 | 0.00 | 99.57 | 0.36 | 3.62 | 92.10 | 3.92 |
| JP06-4 | 35.31 | 0.15 | 0.06 | 0.76 | 30.31 | 0.00 | 0.24 | 33.52 | 0.00 | 100.35 | 0.49 | 2.54 | 94.54 | 2.44 |
| JP06-5 | 34.96 | 0.25 | 0.08 | 1.03 | 29.30 | 0.02 | 0.17 | 33.70 | 0.02 | 99.52 | 0.78 | 3.46 | 91.65 | 4.11 |
| JP06-6 | 35.26 | 0.07 | 0.03 | 9.71 | 21.00 | 0.02 | 0.22 | 33.70 | 0.02 | 100.03 | 0.22 | 32.22 | 64.43 | 3.12 |
| JP06-7 | 34.91 | 0.16 | 0.10 | 8.03 | 22.62 | 0.06 | 0.11 | 33.65 | 0.01 | 99.64 | 0.50 | 26.77 | 69.61 | 3.11 |
| JP06-8 | 34.92 | 0.09 | 0.09 | 1.61 | 29.17 | 0.04 | 0.17 | 33.42 | 0.00 | 99.51 | 0.28 | 5.39 | 91.43 | 2.90 |
| JP06-9 | 33.78 | 0.37 | 0.11 | 1.55 | 27.50 | 0.05 | 0.24 | 32.66 | 0.01 | 96.26 | 1.21 | 5.34 | 88.67 | 4.78 |
| JP06-10 | 35.26 | 0.06 | 0.02 | 5.04 | 25.72 | 0.05 | 0.19 | 33.55 | 0.00 | 99.89 | 0.20 | 16.78 | 80.25 | 2.77 |
| JP06-11 | 35.02 | 0.14 | 0.05 | 0.98 | 29.34 | 0.04 | 0.33 | 33.25 | 0.03 | 99.18 | 0.45 | 3.29 | 92.63 | 3.63 |
| JP08-1 | 35.15 | 0.04 | 0.07 | 3.75 | 26.90 | 0.08 | 0.22 | 33.69 | 0.00 | 99.89 | 0.11 | 12.48 | 83.71 | 3.69 |
| JP08-2 | 34.76 | 0.04 | 0.10 | 1.41 | 28.91 | 0.03 | 0.28 | 33.71 | 0.00 | 99.25 | 0.12 | 4.73 | 90.13 | 5.02 |
| JP08-3 | 34.96 | 0.00 | 0.12 | 0.05 | 31.47 | 0.00 | 0.20 | 33.74 | 0.02 | 100.55 | 0.00 | 0.16 | 96.71 | 3.13 |
| JP08-4 | 34.16 | 0.11 | 0.15 | 0.02 | 30.26 | 0.04 | 0.29 | 33.63 | 0.02 | 98.67 | 0.37 | 0.05 | 93.70 | 5.89 |
| JP08-5 | 35.22 | 0.06 | 0.07 | 4.48 | 25.77 | 0.01 | 0.19 | 33.57 | 0.01 | 99.36 | 0.18 | 14.98 | 81.38 | 3.46 |
| JP08-6 | 35.60 | 0.03 | 0.05 | 0.66 | 30.39 | 0.00 | 0.12 | 33.62 | 0.00 | 100.47 | 0.09 | 2.17 | 95.72 | 2.01 |

## 5. Discussion

The small mountain streams of New Guinea annually transport $1.7 \times 10^9$ t of sediment into the world's oceans [41]. The transported minerals are currently deposited on a narrow shelf and can be found on the edge of the New Guinea Trench [42,43].

The mineral composition of heavy fraction from examined sea sediments strongly depends on the petrology of the Cyclops Mountains [19,29]. The mineralogy of seabed surface sediments of the Jayapura coast correlates with minerals described on shore by [5,21–23,26]. The primary source of contemporary sediments on the Jayapura coast is the Cyclops Mountains, *i.e.*, the metamorphic complex and ophiolite sequence. Heavy minerals from Central Unit sediments are typical for the metamorphic rocks building the core of the Cyclops Mountains. The Eastern and Western Units contain minerals characteristic for ophiolite series such as chromian spinels, olivine, serpentine subgroup, Cr-garnets, magnetite, small amounts of pyroxene and a mixture of hydrated iron (III) oxide-hydroxides (limonite).

### 5.1. Parental Melts and Tectonic Settings

The genetic model of common occurrence of Cr- and Al-spinels in one ophiolite complex is widely discussed in the literature [44–48]. Composition of chromian spinels can explain how magma processes happened in different geodynamic environments. Within the same ophiolite complex, high-Cr and high-Al chromites coexist. It has been observed worldwide, together with bimodal distribution of high-Cr and high-Al chromites. The first group (high-Cr chromites) is placed in the deep mantle section, while the second (high-Al chromites) occurs close to the Moho transition zone. The chemical composition of chromian spinels in the environment of the supra-subduction zone (SSZ) is controlled by the parental melts and by the interaction of melt-rock or melt-melt with previously depleted peridotites. The Cr# can be used as the indicator of mantle depletion. Dick and Bullen (1984) indicate that the high Cr# value correlates with increasing degrees of partial melting [49]. It is believed that parental melts for high-Cr spinels (Cr# > 60) are connected with the mantle sequence from SSZ, while high-Al spinels originate from melts similar to mid-ocean ridge basalt (MORB), which are formed in mid-ocean ridges or in back-arc basins [50,51].

$TiO_2$ contents of the melts in equilibrium with high-Cr spinel were used [44,45,52] to calculate expression the $Al_2O_3$:

$$(Al_2O_3)_{melt} = 5.2253 \cdot \ln(Al_2O_{3\ spinel}) - 1.1232 \tag{1}$$

$$(TiO_2)_{melt} = 1.0897 \cdot \ln(TiO_2)_{spinel} + 0.0892 \tag{2}$$

and for high-Al spinel:

$$(Al_2O_3)_{melt} = 4.1386 \cdot \ln(Al_2O_3)_{spinel} + 2.2828 \tag{3}$$

$$(TiO_2)_{melt} = 0.708 \cdot \ln(TiO_2)_{spinel} + 1.6436 \tag{4}$$

An empirical formula introduced by [53] was used to calculate FeO/MgO melt:

$$\ln(FeO/MgO)_{spinel} = 0.47 - 1.07 Y_{spinel}{}^{Al} + 0.6407 Y_{spinel}{}^{Fe3+} + \ln \cdot (FeO/MgO)_{liquid} \tag{5}$$

where MgO and FeO are in wt%,

$$Y_{spinel}{}^{Fe3+} = Fe^{3+}/(Cr + Al + Fe^{3+}) \tag{6}$$

$$Y_{spinel}{}^{Al+} = Al/(Cr + Al + Fe^{3+}) \tag{7}$$

Calculations show that parental melts (from which chromian spinels crystallized) are composed of Cr-spinels in the following order (wt%): $Al_2O_{3\ melt}$ (12.98–14.89), $TiO_{2\ melt}$ (0.09–0.41), FeO/MgO$_{melt}$ (0.29–0.98), and of Al-spinels (wt%): $Al_2O_{3\ melt}$ (15.02–16.43), $TiO_{2\ melt}$ (0.01–0.73), FeO/MgO$_{melt}$ (0.01–0.45).

The TiO$_2$ vs. Al$_2$O$_3$ diagram (Figure 8) shows that all chromian spinels occur in SSZ and MORB peridotites fields. Therefore, the calculated bimodal split of spinels (Figure 9) can be interpreted as a result of a single magma fractionation, which quickly differentiates in short distance. Chromian spinels composition reflects a vertical zonation due to the fractionation of single batch magma with an initial high-Cr boninitic composition during its ascent. Differentiation occurs in the deep mantle section and above the Moho transition zone. It confirms the thesis given by [5] that the formation of boninites followed during the earliest stages (initial stage) of plate subduction beneath the northern Australian margin. Differentiation of magma took place during magma ascent. The chemical composition of residual peridotites was modified by interaction with different melt (boninite) or fluids.

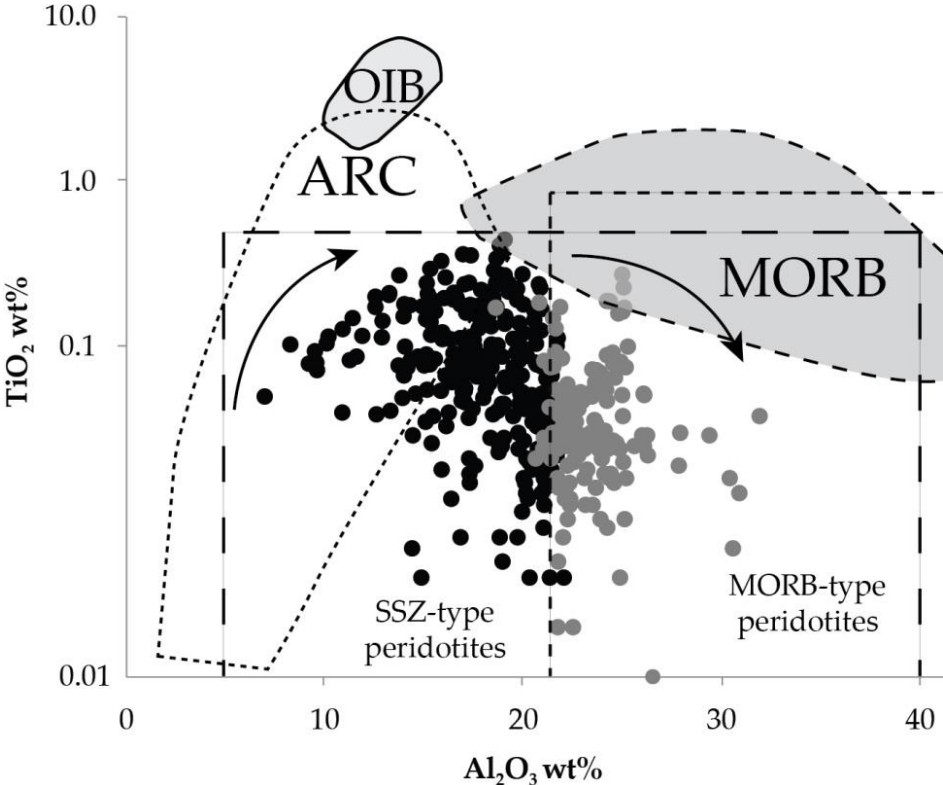

**Figure 9.** The TiO$_2$ vs. Al$_2$O$_3$ tectonic discrimination diagram, modified from [52]. Plots of analyzed samples from Jayapura. Black dots: high-Cr spinels, grey dots: high-Al spinels.

Dependence plots (Figure 10A,B) show that the origin of spinels is tied to an island arc. Chromian spinels were most probably formed in the mantle environment from boninite magma with a very low TiO$_2$ concentration (Figure 10B). Low content of TiO$_2$ (0.30 wt%) is typical for podiform chromites (Figure 10A,B). Data from Figure 9 suggest differentiation of a single melt with an initial composition within the arc field. The increase in Al$_2$O$_3$ content and almost no increase in TiO$_2$ may prove the impact of boninite melt on residual peridotites. This proves melt differentiation towards MORB. The change of Cr# and Mg# indexes shows the gradual chemical depletion of original peridotites (Figure 11A) [54].

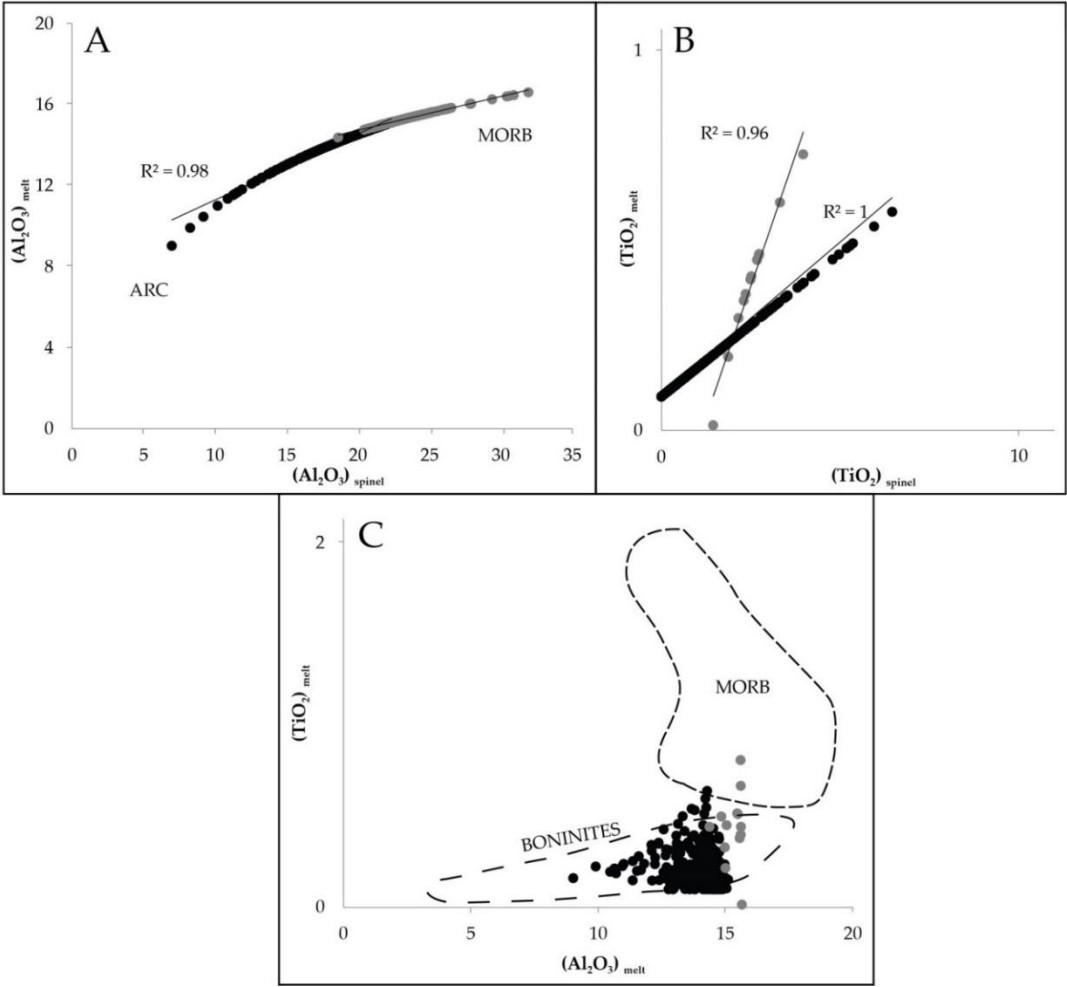

**Figure 10.** (**A**) $Al_2O_3$ melt vs. $Al_2O_3$ spinel, and (**B**) $TiO_2$ melt vs. $TiO_2$ spinel relationships modified from [44,45,52]; (**C**) the melt of $TiO_2$ melt vs. $Al_2O_3$ melt was calculated as in equilibrium with chromite. Plot of analyzed samples from Jayapura. Fields for boninites and mid-ocean ridge basalt (MORB) modified from [55,56]. Black dots: high-Cr spinels, grey dots: high-Al spinels.

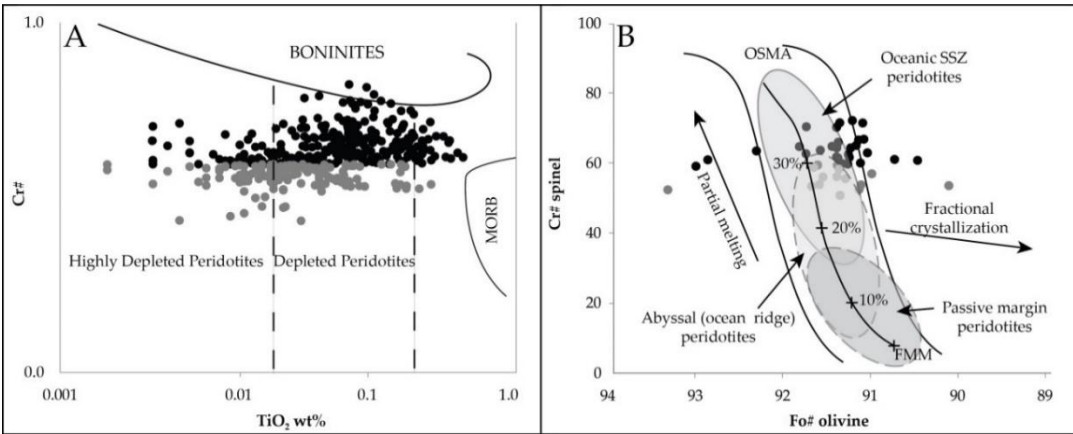

**Figure 11.** Plot of analyzed samples from Jayapura: (**A**) $TiO_2$ vs. Cr# for chromian spinel (modified from [57]); (**B**) Olivine-spinel mantle array $Cr\#_{spinel}$ vs. $Fo\#_{olivine}$ (modified from [56]). Black dots: high-Cr spinels, grey dots: high-Al spinels.

The composition and origin of chromian spinels is very similar to chromitite from Sulawesi reported by [54]. Spinels were generated in the SSZ environment from a single batch magma with initial high-Cr bonititic composition.

A diagram (Figure 11A) indicates the high level of peridotite depletion in the initial melt. It is proved by [5], who established the level of peridotite depletion using REE and trace elements. Geochemical standards of peridotites from the Cyclops Mountains, strongly enriched in LREE and in Zr and Hf, indicate metasomatic peridotites that formed above the subduction zone. A diagram (Figure 11B) shows that partial melting of peridotites in the SSZ environment varies from 25% to 35%. The same value of partial melting (25–35%) was assessed by [5]. Research indicates that peridotites underwent a metasomatic episode connected with a reaction of Ca(?)-boninite melt.

### 5.2. Olivine and Cr-Chromite as Indicators of Rock Origin

Olivine is an important group of minerals used as indicators of rock origin: upper mantle versus mafic and ultramafic magmas [58,59]. An analysis of olivine showed the process of their alteration from unaltered to strongly serpentinized grains. According to [5], olivine from the Cyclops Mountains shows different levels of serpentinization. Olivine from shelf sediments has a high content of Fo (90–92), the magnesium value Mg# is 90–93 and NiO content is 0.30–0.55 ppm with a very low content of Ca (below 500 ppm). A correlation diagram [59] between Ca vs. forsterite (Figure 12A) places obtained results in the field of mantle olivine. There is no clear correlation between $NiO_2$ wt% content with forsterite (Figure 12B). This suggests that olivine may have crystallized from mantle magma, thus forming cumulates of ophiolite rocks of the Cyclops Mountains. Olivine replaced by Cr-garnets is found in samples. It may indicate metasomatic changes in peridotites. The existence of Cr-garnet with serpentinized olivine most probably relates to interaction of Ca?-boninite melt with peridotites. A similar mineral phase was described from a chromitite layer in the Jijal dunites complex in NW Pakistan. Garnet was formed as a result of retrogression in the greenschist facies [60].

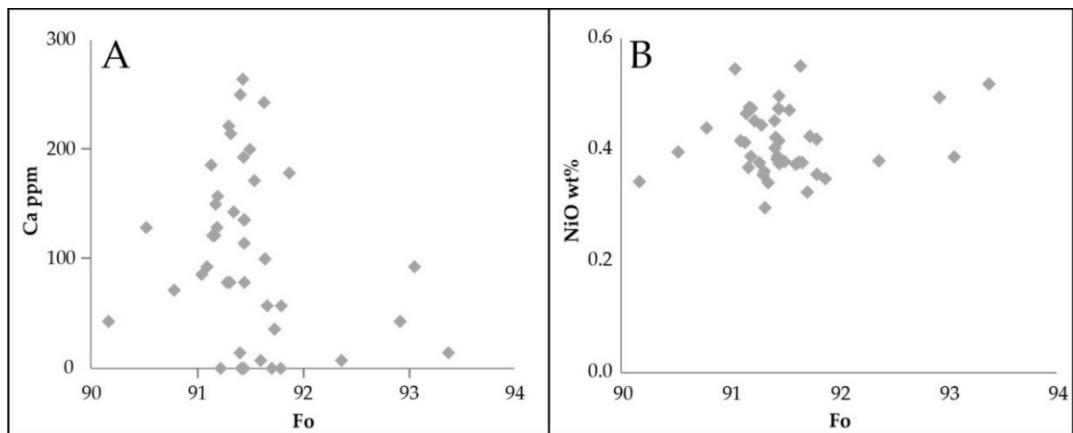

**Figure 12.** Variation of Ca ppm (**A**), $NiO_2$ wt% (**B**) as a function of forsterite molar % in the olivine from marine sediments.

Olivine originates from dunites and harzburgites of the Cyclops Mountains. A similar Mg# value is presented by [5]. They present a high magnesium value (Mg# 90–92) and high content of NiO (0.38–0.45 wt%) [5]. The geochemical similarity of olivine from the Cyclops Mountains massif indicates its similar source.

Chromite ore body has not been discovered in the Cyclops ophiolite. As many authors indicated [61–63], the generation of boninite magmas can result from melting from the plate in the amphibolite facies and depleted MORB melting below 50 km. Lago et al. (1982) describes the chromite podiform mechanism in which mafic magmas originate from the lower parts of the crust, migrate upward through narrow channels, and precipitate chromites from the magma chamber of

the upper mantle tectonite [64]. A similar mechanism was present during the formation of Cyclops ophiolite. As a result of strong convection currents, rapid precipitation of chromite spinels occurred in the magma chamber that did not agglomerate into large ore bodies. Concordantly, no large magma bodies were detected in the Cyclops ophiolite, only narrow, discontinuous chromite layers [5].

## 5.3. Platinum Group Metals

PGMs have been found in several locations in the Indonesian Archipelago: (1) chromite deposits in the South and Southeast of Sulawesi [54]; (2) Alaskan-type ultramafic intrusions in the Kalimantan [65]; (3) skarn deposits in Sumatra [66] and (4) placer deposits in South Kalimantan [67]. Studies of rocks from the Cyclops Mountains conducted by [68] showed the presence of platinum group metals in serpentine. There are no other data to confirm it. At the end of the 1980s, the PT Sentani Maju Minerals company conducted exploration research in the Tanahmerah region, focused on the platinum group metals and, on the basis of 19 samples from shallow boreholes, platinum content was assessed at 35 ppb, while palladium only reached 10 ppb [23,69]. The promising presence of PGM in the samples [23,69] shows the need for further detailed studies of the nature, source and form of PGM occurrences.

Ophiolite rocks contain a considerable variety of precious metals. This is due to fractionation and magma differentiation processes. The geochemistry of PGEs in minerals and alloys closely relates to the presence of relative sulfur saturation, fugacity $f(S_2)$ and temperature of parental melts forming mantle-hosted chromites in the supra-subduction zone [70–73]. The PGE amount is determined through fractionation during crystallization of the parent melt and partial melting of the mantle source. PGM are lithologically associated with serpentinized harzburgites, dunites and lherzholites [70]. Usually, PGM occur in micrometric forms in chromites or are connected to rims of larger base-metal sulfides, *i.e.*, pentlandite $(Fe,Ni)_9S_8$, chalcopyrite $(CuFeS_2)$ or awaruite $(Ni_3Fe)$ [70]. Generally, podiform chromitites (Cr-high) crystallized from boninitic melts in SSZ settings have high IPGE/PPGE ratios, unlike high-Al chromite which usually has a lower PGM content [73].

Pt content (up to 9 ppb) was determined using ICP-MS analyses. We observed one post-magmatic generation of PGM. The secondary phase consists of very tiny veins of Os-Cu assemblage. We believe that the phase formed in low-temperature processes during exsolution of chromian spinels.

## 5.4. Necessity of Future Deposit Research

Chromite deposits of podiform type and Cr placer deposits are known in Indonesia from many locations, such as: South Kalimantan, Sulawesi, Maluku, Halmahera, Gebe, Gag, Waigeo, and Papua. That is why further research of grassroots and late stage type should be conducted in Indonesia. The form and resources of placer chromite deposits depend on the coastal environment that is characterized by dynamic deposit formation processes leading to heavy mineral concentration. Mechanical and chemical weathering processes, hydraulic sorting, abrasion and dissolution may lead to high concentrations of heavy minerals in sea deposits.

Sediment composition is a major property of many seabed substrates. They come from coastal (terrigenous sediments), in situ (biogenous sediments produced by organisms and hydrogenous sediments) and ex-situ (transferred by currents and waves). Contemporary sediments are poorly consolidated. Medium and fine grained sediments have varying content of heavy minerals fraction [19].

The heavy fraction of these sediments is composed of chromian spinels, olivine, ilmenites, amphiboles, magnetites, pyroxenes and minerals of serpentine group and in minor amounts of sulphides. Heavy mineral grains are present in characteristic paragenesis and textures within irregular veins such as lamellae, etc. Textures and mineral paragenesis impact technology and the processing cost of marine sediments.

However, the content of $Cr_2O_3$ is below economic grade in the samples investigated by the authors. Land sediments around the Cyclops Mountains have been analyzed since the 1920s. In 1977, the Indonesian Geological Survey, together with the Australian Bureau of Mineral Resources, Geology and Geophysics (BMR), conducted geological prospection focusing on recognition of deposit potential

in the Cyclops Mountains. In a final report prepared in 1979 by Australian geologists (BMR), Jayapura Regency was indicated as a region with negative prognosis for economic deposits of nickel, cobalt and chromium [74]. The Cyclops Mountains have been designated as a nature reserve park since 1978 due to the unique and varied wealth of endemic species of fauna and flora [75]. Therefore, even if economic deposits were recognized, mining activities around the Cyclops Mountains massif would be forbidden. However, the area beyond the nature park—beach sediments and shallow water sediments—is geologically interesting and should to be examined. Results of their detailed studies have contributed to geological and chemical recognition [19]. Therefore, future research should focus on shallow offshore sediments. This will allow prognoses of deposit prospects.

Results presented in this paper did not reveal deposit concentrations of Ti, Ni, Co, Cr, Au along the Jayapura coast. However, the confirmed slight increase in content of Cu-, Zn- and Pb- in heavy mineral fraction (Appendix A) is probably associated with identified sulphides (0.1–0.3 wt%). Positive anomaly of Ag (up to 5000 ppb) and W (up to 3130.3 ppm) in sediments requires further detailed studies of the region.

## 6. Conclusions

1. On the basis of geochemical and mineralogical characterization, marine sediments from the Jayapura coast have been divided into three units: Eastern, Central and Western. The analyzed marine sediments contain heavy minerals (on average) in the Eastern Unit (1.93 wt%), Central Unit (26.47 wt%) and Western Unit (24.92 wt%).

2. Heavy mineral fraction separated from marine sediments along the coast of the Jayapura Regency (New Guinea) is polymineralic and mainly consists of high-Al and high-Cr spinels, chromian andradite, Mg-olivine and minerals from serpentine group (lizardite, antigorite) and minor magnetite, as well as a mixture of iron (III) oxyhydroxides (limonite).

3. The heavy mineral fraction contains increased content of W (up to 3130.3 ppm) and Ag (up to 5000 ppb). The positive anomaly of Ag and W requires further detailed studies.

4. On the basis of geochemical analyses, the source of heavy minerals was determined to be the ophiolite rock sequence of the Cyclops Mountains.

5. The chemical composition of chromian spinels shows that they originate from depleted peridotites generated in the SSZ environment. Parental melt evolved toward magma with geochemical parameters similar to MORB. The geochemical factor indicates that the origin of spinels is associated with an island arc. Chromian spinels were most probably formed in the mantle environment from boninite lava with a very low $TiO_2$ concentration. The peridotites underwent intense metasomatic processes by Ca?-boninite melt. As a result of inflow of Ca-rich solutions, olivine was altered and replaced by chromian andradite.

6. The results did not reveal deposit concentrations of such metals as Ti, Ni, Co, Cr, Au in investigated marine sediments along the Jayapura coast.

7. The sediments contain sulphides (0.1–0.3 wt%) such as pyrite (dominating), chalcopyrite, covelline and bornite. Slightly increased content of Cu (up to 58.90 ppm), Zn (up to 118 ppm) and Pb (up to 7.40 ppm) in heavy mineral fraction may suggest that they are present in the sulphides. Further study is required to confirm this thesis.

**Author Contributions:** Conceptualization, K.Z. and K.S.; methodology, K.Z.; formal analysis, K.Z.; investigation, K.Z., resources, K.Z.; data curation, K.Z. and K.S.; writing—original draft preparation, K.Z.; writing—review and editing, K.Z. and K.S.; visualization, K.Z.; supervision, K.Z., K.S., I.G.; project administration, K.Z., K.S. and I.G. All authors have read and agreed to the published version of the manuscript.

**Funding:** This research was co-founded by statutory funds from the Polish Geological Institute-National Research Institute (No 62.9012.1959.00.0) and the University of Warsaw, Faculty of Geology.

**Acknowledgments:** The authors would like to thank anonymous reviewers for their detailed opinion, as well as remarks and suggestions that allowed us to improve the final version of the manuscript.

**Conflicts of Interest:** The authors declare no conflict of interest.

# Appendix A

**Table A1.** Chemical compositions of heavy mineral fractions. ICP-MS/ES analysis.

| Area | Sample | Fraction | Cu | Cr | Ni | Pb | Zn | Co | Cd | Sb | Bi | Li | Mo | Sn | V | W | Zr | Nb | Hf | Th | U | Hg | Ag | Au | Pt | Pd |
|---|---|---|---|---|---|---|---|---|---|---|---|---|---|---|---|---|---|---|---|---|---|---|---|---|---|---|
| | | | | | | | | | | | | ppm | | | | | | | | | | | | ppb | | |
| Eastern Unit | | <0.10 mm | 1.97 | 170.6 | 2522.7 | 0.39 | 27.8 | 110.3 | 0.01 | <0.02 | 0.72 | 3.8 | 0.18 | <0.1 | 6 | 57.2 | <0.1 | <0.02 | <0.02 | <0.1 | <0.1 | <5 | 7 | <0.2 | 5 | <10 |
| | JP 07 | <0.25 mm | 2.26 | 216.6 | 2261.6 | 0.24 | 33.7 | 99.6 | 0.12 | 0.02 | <0.02 | 262.8 | 0.43 | <0.1 | 6 | 48.2 | 0.05 | <0.02 | <0.02 | <0.1 | <0.1 | <5 | 2328 | 0.6 | 3 | <10 |
| | | <0.50 mm | 3.03 | 190.2 | 2217.9 | 0.08 | 31.0 | 97.0 | 0.09 | <0.02 | <0.02 | 230.2 | 0.18 | 0.2 | 5 | 72.1 | 0.03 | <0.02 | <0.02 | <0.1 | <0.1 | <5 | 2150 | <0.2 | <2 | <10 |
| | JP 08 | <0.10 mm | 2.31 | 167.0 | 2548.1 | 0.56 | 29.9 | 120.3 | 0.01 | 0.03 | <0.02 | 4.5 | 0.68 | <0.1 | 7 | 99.4 | 0.1 | <0.02 | <0.02 | <0.1 | <0.1 | <5 | 43 | 0.4 | <2 | <10 |
| | | <0.25 mm | 2.53 | 160.7 | 2454.8 | 2.36 | 27.9 | 113.8 | <0.01 | 0.04 | 0.03 | 2.4 | 0.19 | <0.1 | 7 | 48.8 | 0.2 | <0.02 | <0.02 | <0.1 | <0.1 | 16 | 20 | <0.2 | <2 | <10 |
| | JP 09 | <0.10 mm | 2.40 | 666.0 | 2116.7 | 0.99 | 28.1 | 94.4 | 0.01 | 0.05 | <0.02 | 11.9 | 0.25 | <0.1 | 17 | >100 | 0.1 | <0.02 | <0.02 | <0.1 | 0.2 | <5 | 24 | 0.8 | 3 | <10 |
| | | <0.25 mm | 2.63 | 179.0 | 2406.6 | 0.51 | 28.5 | 108.6 | <0.01 | 0.05 | <0.02 | 3.9 | 0.22 | <0.1 | 9 | >100 | 0.3 | <0.02 | <0.02 | <0.1 | 0.1 | <5 | 19 | <0.2 | 3 | <10 |
| | JP 10 | <0.10 mm | 3.98 | 452.6 | 2364.2 | 0.66 | 32.2 | 116.5 | <0.01 | 0.04 | <0.02 | 4.1 | 0.16 | <0.1 | 16 | >100 | 0.2 | <0.02 | <0.02 | <0.1 | <0.1 | <5 | 27 | 0.4 | <2 | <10 |
| | | <0.25 mm | 4.26 | 591.4 | 2549.0 | 0.57 | 36.2 | 135.9 | 0.01 | 0.05 | <0.02 | 9.8 | 0.37 | <0.1 | 22 | >100 | 0.3 | <0.02 | <0.02 | <0.1 | 0.2 | <5 | 19 | 1.1 | <2 | <10 |
| | | <0.50 mm | 3.39 | 580.7 | 2256.6 | 0.51 | 29.2 | 117.5 | 0.01 | 0.04 | <0.02 | 6.5 | 0.25 | <0.1 | 18 | >100 | 0.1 | <0.02 | <0.02 | <0.1 | <0.1 | <5 | 12 | 1.1 | <2 | <10 |
| Central Unit | JP 13 | <0.10 mm | 16.30 | 7544.0 | 440.0 | 3.30 | 118.0 | 60.1 | 0.80 | 0.20 | <0.02 | 250.9 | 0.30 | 1.1 | 383 | >100 | 8.0 | 5.40 | 0.40 | 0.9 | 0.4 | <5 | 4200 | <0.2 | <2 | <10 |
| | JP 14 | <0.10 mm | 43.26 | 34.10 | 26.5 | 0.59 | 36.2 | 14.0 | 0.23 | 0.08 | <0.02 | 468.3 | 0.15 | 0.1 | 165 | 81.3 | 2.6 | 0.12 | 0.11 | 0.1 | 0.1 | 23 | 4899 | <0.2 | 3 | <10 |
| | JP 29 | <0.50 mm | 21.00 | 316.0 | 74.6 | 7.40 | 69.0 | 21.6 | 0.90 | <0.02 | <0.02 | 387.3 | 0.10 | 1.1 | 247 | 20.5 | 6.2 | 0.90 | 0.30 | 1.1 | 0.7 | <5 | 5700 | <0.2 | <2 | <10 |
| | JP 32 | <0.10 mm | 36.60 | 240.0 | 90.9 | 3.50 | 112.0 | 22.9 | 1.90 | <0.02 | <0.02 | 866.6 | 0.10 | 0.4 | 140 | >100 | 3.3 | 0.70 | 0.20 | 0.4 | 0.3 | <5 | 12,200 | <0.2 | <2 | <10 |
| | | <0.25 mm | 58.90 | 279.0 | 90.8 | 5.20 | 116.0 | 25.1 | 2.0 | <0.02 | <0.02 | 793.6 | 0.20 | 0.6 | 183 | 32.0 | 4.4 | 0.70 | 0.20 | 0.6 | 0.5 | <5 | 12,100 | <0.2 | <2 | <10 |
| Western Unit | JP 36 | <0.10 mm | 26.07 | 117.3 | 35.2 | 1.01 | 36.9 | 12.7 | 0.18 | <0.02 | <0.02 | 392.0 | 0.09 | 0.2 | 59 | 19.8 | 0.5 | <0.02 | 0.02 | 0.1 | <0.1 | 29 | 4090 | 0.4 | 4 | <10 |
| | | <0.25 mm | 19.05 | 68.7 | 21.5 | 0.71 | 32.1 | 8.1 | 0.23 | <0.02 | <0.02 | 558.2 | 0.07 | <0.1 | 27 | 29.7 | 0.4 | <0.02 | 0.02 | <0.1 | <0.1 | <5 | 5390 | <0.2 | <2 | <10 |
| | JP 41 | <0.10 mm | 10.05 | 45.20 | 17.1 | 1.72 | 45.3 | 6.7 | 0.23 | 0.04 | <0.02 | 555.8 | 0.11 | 0.3 | 67 | 17.4 | 1.5 | 0.04 | 0.07 | 0.3 | 0.5 | 19 | 5248 | <0.2 | <2 | <10 |
| | | <0.25 mm | 12.98 | 30.0 | 33.2 | 1.24 | 60.5 | 10.1 | 0.21 | 0.04 | <0.02 | 513.0 | 0.10 | 0.2 | 51 | 29.7 | 0.8 | 0.02 | 0.05 | 0.2 | 0.5 | <5 | 4419 | <0.2 | <2 | <10 |
| | | <0.50 mm | 5.88 | 19.0 | 21.2 | 1.47 | 23.1 | 3.8 | 0.20 | 0.04 | <0.02 | 513.0 | 0.09 | 0.2 | 38 | 34.7 | 1.0 | 0.05 | 0.06 | 0.3 | 0.5 | <5 | 4574 | <0.2 | <2 | <10 |
| | JP 43 | <0.50 mm | 12.41 | 20.7 | 7.7 | 2.52 | 45.2 | 6.8 | 0.20 | 0.07 | <0.02 | 564.0 | 0.12 | 0.1 | 44 | 36.9 | 0.7 | 0.03 | 0.03 | 0.1 | 0.3 | 9 | 5530 | <0.2 | <2 | <10 |
| | JP 49 | <0.10 mm | 3.12 | 297.0 | 2697.6 | 0.49 | 30.9 | 112.4 | <0.01 | 0.02 | 0.03 | 5.9 | 0.12 | <0.1 | 11 | >100 | 0.1 | <0.02 | <0.02 | <0.1 | <0.1 | <5 | 8 | <0.2 | 5 | <10 |
| | | <0.25 mm | 3.05 | 211.2 | 2428.8 | 0.39 | 26.8 | 101.1 | <0.01 | <0.02 | <0.02 | 5.8 | 0.08 | <0.1 | 8 | 96.8 | <0.1 | <0.02 | <0.02 | <0.1 | <0.1 | <5 | 6 | <0.2 | 4 | <10 |
| | | <0.50 mm | 3.60 | 107.1 | 2134.5 | 0.23 | 22.3 | 104.8 | 0.01 | <0.02 | <0.02 | 3.6 | 0.13 | <0.1 | 5 | 83.6 | <0.1 | <0.02 | <0.02 | <0.1 | <0.1 | <5 | 7 | <0.2 | 3 | <10 |

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
