# Peer review of "The Cyclops Ophiolite as a Source of High-Cr Spinels from Marine Sediments on the Jayapura Regency Coast (New Guinea, Indonesia)"

_minerals, doi:10.3390/min10090735_

Round 1

Reviewer 1 Report

Please see the attached file for detailed comments. 

Author Response

Dear Reviewer,

Thank you very much for fruitfull remarks and comments.

I enclose the file with answers.

Kind regards,

prof. Krzysztof (Chris) Szamałek

Reviewer 2 Report

Dear Authors

The manuscript I reviewed deals with the study of heavy minerals concentrates from beach and shelf sediment on the coast of the Cyclops ophiolite mountain range. I am sorry but I have to say that the work has serious flaws.

The first two paragraphs of Introduction are a very broad description of ophiolites from Southeastern Asia and New Guinea that is not clear at all, it is impossible reading to figure out the evolution of this area and the formation of several opholite types, a figure with map could also help.

No areal distribution of the sampling location is given, a general subdivision in three parts based on some previous work is not clear and the results concern only two of these three parts. No data are provided on the composition and grain size distribution of the sediment sampled, no info are provided on the weight fraction of the heavy mineral in respect to the whole sediment, the tables report different grain size classes for different samples without any explanation.

The main goal of the manuscript seems to be the study of peridotite minerals and reconstruction of genesis and alteration of the original rocks, but this can be much better done sampling the ophiolite and the article by Monnier et al [5] in the text presents such kind of data. So data presented here are of lower significance as no texture or lithology is preserved in single grains of sediment. The discussion concerning the origin of the spinels has also some flaws and anyway can be much better done with data from the ophiolite and not from the sediments, for example you do not know if your chromite grains are from chromitite or from host peridotite.

The manuscript focuses then on PGE but the contents of these metals (really only Pt and Pd) according to whole rock analyses is very low and not meaningful at all. By the way it is well known that chromitites are depleted in Pd and Pt and enriched in IPGE, so must analyse the sediment for all the 6 PGE.

On the other hand some interesting data such as the quite high concentration of Ag in the heavy concentrate are not discussed at all, what is mineralogy of Ag in your samples? And also the mineralogy of the high Ni content is not discussed, is this Ni in olivine or in sulphides (some of which were detected but not analysed).

I must reject this manuscript and suggest you to make further analyses and focus on other interesting results such as Ag and Ni positive anomalies in the sediments, you should also discuss the history of the minerals from the mother rock in the ophiolite to the sediment.

Some more specific comments can be found in annotated manuscript.

Author Response

Dear Reviewer,

Thank you very much for your fruitfull remarks.

I enclose the file with answers.

Kind regards,

prof. Krzysztof (Chris) Szamałek

Reviewer 3 Report

The manuscript reports a great number of mineralogical data on sediments of New Guinea, Indonesia, poorly studied so far. The Authors report on the presence of chromite and silicates providing  a good description and proper characterization based on electron microprobe analyses. Whole rock analyses are also reported to verify the presence of certain metals. Unfortunately I was not able to check most of the tables because their bad format. I strongly suggest to the authors to reorganize properly them. The authors used the collected data, especially the composition of chromite, to understand the provenance of the studied samples. For me it is suitable for publication in Minerals after minor corrections. Here are some suggestions that I hope will help the Authors to improve their manuscript.

I was really surprise that the composition of the chromite is very similar to those reported from Sulawesi chromitite by Federica Zaccarini, Arifudin Idrus and Giorgio Garuti (Chromite Composition and Accessory Minerals in Chromitites from Sulawesi, Indonesia: Their Genetic Significance, Minerals 2016, 6(2), 46; https://doi.org/10.3390/min6020046). I suggest to the authors to read this manuscript and maybe to make a comparison.

Regarding the composition of the analysed chromite and the presented graphs, most of the data plot in the field of SSZ chromitite but the parental magma shows also a MORB affinity. The Authors should explain better this point, may be they analyses spinels derived from different type of ophiolites or may be their composition reflects  a vertical zonation due to the fractionation of a single batch magma with an initial boninitic composition during its ascent, in a supra-subduction zone as suggested by Zaccarini et al. 2016 for the Sulawesi chromitites.

Olivine composition can also be used as petrogenetic indicator. In particular Li, C.; Thakurta, J.; Ripley, E.M. (Low-Ca contents and kink-banded textures are not unique to mantle olivine. Evidence from the Duke Island Complex, Alaska. Mineral. Petrol. 2012, 104, 147–153) have suggested to use Ca content  of olivine to define its mantle origin, I suggest to the Authors to cite this paper and eventuali to add the binary diagram Ca vs Fo in their manuscript.

The Authors report whole rock analyses for Pd and Pt. Are the other 4 PGE, i.e. Os, Ir, Ru and Rh below detection limits or they were not analysed? Please explain this point. The chapter on PGE metals is not very informative. There is huge bibliography showing that the PGE in the ophiolitic chromitite are carried in specific phases and are not hosted in the spinel lattice as can happen in few komatiite. The authors should focussed this chapter on the PGE distribution and mineralogy of ophiolitic chromitites that are generally enriched in IPGE compared to PPGE. To find  platinum group mineral (PGM) in podiform chromitite is not an easy task, since they mostly occur as tiny and rare phases. Therefore I guess that to find them in the sediments derived from podiform chromitite is a quite complicate target. In this respect the authors should explain better which fraction was analysed for PGE and PGM. Generally they occur in the very fine fraction. The presence of PGM in Indonesian  area was reported and discussed by several authors (Stumpfl, E.F.; Clark, A.M. Electron-probe microanalysis of glod-platinoid concentrations from Southeast Borneo. Trans. Inst. Min. Metall. 1965, 74, 933–946. Burgath, K.P.; Mohr, M. Chromitites and platinum-group minerals in the Meratus-Bobaris ophiolite zone, southeast Borneo. Metallog. Basic Ultrabasic Rocks 1986, 1986, 333–349, Burgath, K.P. Platinum-Group Minerals Southeast Kalimantan; Prichard, H.M., Potts, P.J., Bowles, J.F.W., Cribb, S.J., Eds.; Springer: Rotterdam, The Netherlands, 1988; pp. 383–403., Zientek, M.L.; Pardiartob, B.; Simandjuntakb, H.R.W.; Wikramac, A.; Oscarsond, R.L.; Meiere, A.L.; Carlsone, R.R. Placer and lode platinum-group minerals in south Kalimantan, Indonesia: Evidence for derivation from Alaskan-type ultramafic intrusions. Aust. J. Earth Sci. 1992, 39, 405–417, Bowles, J.F.W.; Cameron, N.R.; Beddoe-Stephens, B.; Young, R.D. Alluvial gold, platinum, osmium-iridium, copper-zinc and copper-tin alloys from Sumatra—Their composition and genesis. Inst. Min. Metall. Trans. 1984, 93, B23–B30, Stumpfl, F.E.; Tarkian, M. Vincentite, a new palladium mineral from south-east Borneo. Mineral. Mag. 1974, 39, 525–527).

Best regards.

Author Response

Dear Reviewer,

we enclosed our answers to you.

Authors

Round 2

Reviewer 2 Report

Dear Authors

I appreciate the extensive effort you have done to change the manuscript according to my suggestions and it was really improved, but all this was done only on the annotate manuscripts I attached in my review where minor issues and problems with English language were addressed. So you improved introduction, analytical methods description, data presentation, tables and figures and English language but did not touch what were the main flaws of the manuscript that I reported in "comments and suggestions" box. To cancel some discussion about PGE content and to state that anomalous contents of Ag and W require further investigation is not enough, I think you must deal with this further investigation, check where this Ag and W are contained and discuss their possible origin and their economic interest. Also discussion on chromite genesis is not novel and does not add much to what is known in literature.

Author Response

Dear Reviewer

we enclose our answers to you.

Authors
